# Conformational changes in intact dengue virus reveal serotype-specific expansion

Xin-Xiang Lim[1,*], Arun Chandramohan[1,*], Xin Ying Elisa Lim[2], Nirmalya Bag[1], Kamal Kant Sharma[1], Melissa Wirawan[2], Thorsten Wohland[1,3,4], Shee-Mei Lok[1,2] & Ganesh S. Anand[1]

Dengue virus serotype 2 (DENV2) alone undergoes structural expansion at 37 °C (associated with host entry), despite high sequence and structural homology among the four known serotypes. The basis for this differential expansion across strains and serotypes is unknown and necessitates mapping of the dynamics of dengue whole viral particles to describe their coordinated motions and conformational changes when exposed to host-like environments. Here we capture the dynamics of intact viral particles of two serotypes, DENV1 and DENV2, by amide hydrogen/deuterium exchange mass spectrometry (HDXMS) and time resolved Förster Resonance Energy Transfer. Our results show temperature-dependent dynamics hotspots on DENV2 and DENV1 particles with DENV1 showing expansion at 40 °C but not at 37 °C. HDXMS measurement of virion dynamics in solution offers a powerful approach to identify potential epitopes, map virus-antibody complex structure and dynamics, and test effects of multiple host-specific perturbations on viruses and virus-antibody complexes.

[1] Department of Biological Sciences, National University of Singapore, 14 Science Drive 4, Singapore 117543, Singapore. [2] Duke-National University of Singapore Graduate Medical School, 8 College Road, Singapore 169857, Singapore. [3] Department of Chemistry, National University of Singapore, 14 Science Drive 4, Singapore 117543, Singapore. [4] Centre for Bioimaging Sciences, National University of Singapore, 14 Science Drive 4, Singapore 117557, Singapore. * These authors contributed equally to this work. Correspondence and requests for materials should be addressed to G.S.A. (email: dbsgsa@nus.edu.sg).

Dengue fever, caused by the dengue virus (DENV), is a major global challenge with 390 million estimated infections annually[1]. There are four distinct serotypes of DENV and infection by one serotype does not confer immunity against heterologous serotypes[2] but may instead cause a more severe and life-threatening form of dengue infection through antibody-dependent enhancement[3]. Although there is a commercially available vaccine, its efficacy is poor[4] and there are no alternative therapeutics[3–6].

DENV is a member of the *Flaviviridae* family and is transmitted to humans via the bite of a mosquito vector (*Aedes sp.* of mosquitoes)[7]. The mature viral particle consists of a single stranded RNA genome, a lipid bilayer membrane and a structural proteome consisting of three proteins: capsid (C), envelope (E) and membrane (M) proteins[8]. The E- and M-proteins form the outermost viral shell with 180 copies arranged in icosahedral symmetry[9]. The amphipathic stem and transmembrane helices of the E- and M-proteins help anchor them to the lipid bilayer[10] (Fig. 1). C-protein is present inside the viral particle and complexed with viral genomic RNA[11].

Through its life cycle, the viral particle is subjected to various perturbations including temperature, pH and host–protein interactions, among others[12]. Of these, temperature represents one of the important triggers in initiating the infectious phase of the DENV virus upon host cell entry. An increase in temperature during vector (28 °C) to host (37 °C) transmission has been shown to trigger variable degrees of structural transitions across all four dengue serotypes and is most prominent in strains of DENV2, as smooth to rough surface transitions[13,14]. However, serotypes DENV1, 3 and 4, show no observable changes between these temperatures from cryo-EM studies (Fig. 1)[15–17]. These differences in temperature-induced structural transitions are intriguing due to the high sequence similarity (>60%) in their scaffold E-proteins[13–16,18].

Although cryo-EM structures of viral particles describe the viral envelope and symmetry[10,19] at 28 and 37 °C, they only represent a single stable endpoint state and offer limited predictive insights into its large-scale temperature and other host-specific perturbation-dependent transitions[12]. The low resolution (13 Å) of the expanded DENV2 (New Guinea-C (NGC)) structure at 37 °C (ref. 13) precludes high resolution identification and mapping of the expanded state within the E- and M-proteins, while the moderately high resolution structure (4.5 Å) of DENV1 (PVP159) at 28 °C (and 37 °C) does not permit identification of localized domain movements within E- and M-proteins between these temperatures. The high sequence and structural homology of DENV1 and DENV2 at 28 °C does not provide any apparent correlation between sequence/structural assembly[13,14] and the observed differential temperature-dependent expansion in DENV2 and DENV1 in solution. This necessitates a thermodynamic description of the whole viral particle in solution at separate temperatures.

Amide hydrogen/deuterium exchange mass spectrometry (HDXMS) captures thermodynamic transitions of proteins by measuring the increment in mass when backbone amide hydrogens exchange with solvent deuterium[20,21]. The extent of HDX is dependent on hydrogen bonding and solvent accessibility of the protein[22,23], without the limitation of size/quaternary assembly[24], at the seconds and slower time scales[22]. In order to describe the differences in temperature-induced expansion, we used DENV1 (PVP 159) and DENV2 (NGC) for carrying out HDXMS on intact viral particles at 28, 37 and 40 °C. DENV1 and DENV2 in the rest of the manuscript refer specifically to PVP159 and NGC strains, respectively. This enabled mapping of temperature-dependent changes by examining these virus particles in native environments as a composite macromolecular assembly rather than as a sum of tertiary structures of individual components.

We next tracked the expansion process in DENV using time resolved Förster Resonance Energy Transfer (TR-FRET) to determine the temperature threshold for the transition and to monitor the cooperativity of temperature-dependent expansion. Dual fluorescence labelled (E-protein and lipid bilayer) DENV particles were used as TR-FRET probes across a temperature range from 25 to 40 °C to track their expansion.

Here we show that DENV2, at 37 °C, shows temperature-specific differences due to specific assemblies of E-protein on the viral particle and highlight the importance of quaternary contacts in virions. However, DENV1 showed no expansion at 37 °C and exhibited a different expansion profile at 40 °C, with a different set of temperature-specific loci. Further, TR-FRET measurements revealed a transition in temperature-dependence of expansion in DENV2 from 25 to 37 °C, while DENV1 expansion occurred only at 40 °C. Thus a combination of HDXMS and TR-FRET indicate the importance of viral assembly and conformational dynamics to explain differential host-specific responses in structurally similar viral strains and offer powerful insights into viral dynamics and temperature-dependent expansion of DENV2 and DENV1.

## Results

**Amide hydrogen deuterium exchange of whole dengue particles.** In order to capture temperature-induced changes in the two dengue serotypes: Purified DENV1 (PVP 159) and DENV2 (NGC), intact, infectious viral particles were equilibrated at 28, 37 or 40 °C as specified, for 30 min and deuterium exchange was then initiated by 10-fold dilution in deuterated buffer (aqueous buffer reconstituted in $D_2O$) to result in a final deuterium oxide concentration of 90% (refer methods, Supplementary Note 1). Deuterium exchange was carried out for 1 min and the exchange reaction was quenched by rapidly lowering $pH_{read}$ to 2.5 followed by complete proteolysis by online pepsin, chromatographic separation and mass over charge ($m/z$) determination of all pepsin fragmented peptides as described in methods. Sequencing of C, E and M-peptides yielded sequence coverages of 20, 77.2 and 56% for each of the constituent proteins, respectively, from DENV2 and 28, 71.7 and 45.3% for each of the constituent proteins, respectively, from DENV1 (Supplementary Fig. 2a,b). Deuterium exchange was calculated by subtracting centroid masses of deuterium exchanged peptides for each time point from that of undeuterated peptides. Isotopic mass envelopes of selected peptides are shown in Fig. 2a. Mass shifts provided a readout of the average numbers of deuterons exchanged at a given time point of exchange. The relative deuterium exchange for each pepsin fragment peptide was calculated as a ratio of observed exchange to maximum exchangeable amide hydrogens. Mass spectral envelopes were typical of EX2 deuterium exchange kinetics[25]. A relative fractional deuterium uptake (RFU) plot showed a protein-wide relative exchange overview for all DENV proteins (Fig. 2b,c, Supplementary Fig. 4a,b).

Our initial experiments showed no increases in deuterium exchange for time points from 1 min up to 60 min (Supplementary Fig. 3), suggesting that 1 min of deuterium exchange offered an optimal window for capturing temperature-specific perturbations on DENV2 particles (Supplementary Note 1, Supplementary Fig. 3a,b). Hence deuterium exchange at 1 min for both DENV2 and DENV1 particles across all pepsin fragmentation peptides allowed effective mapping of temperature-specific effects on intact viral particles at all temperatures tested. It should be noted that the 180 copies of the E-protein are organized with icosahedral symmetry resulting in 12 five-fold

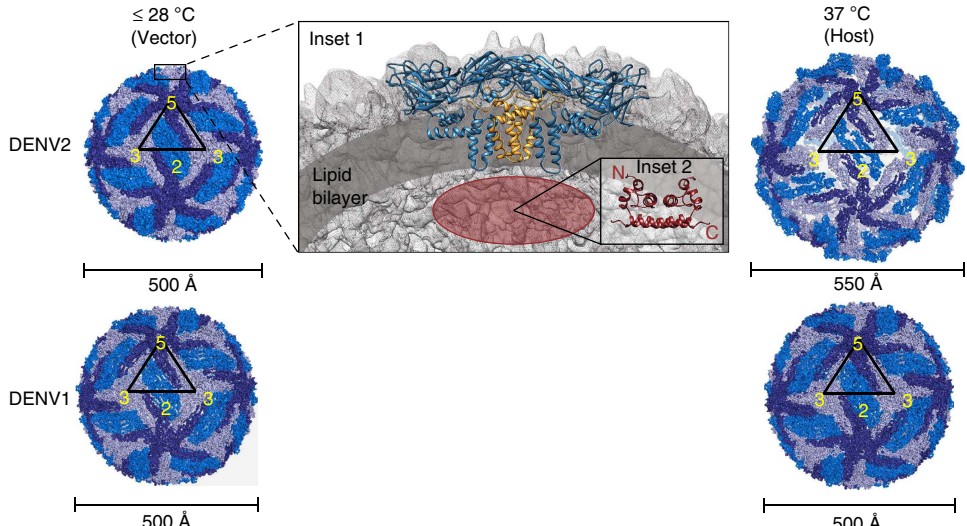

**Figure 1 | Temperature-dependent changes in DENV1 and 2 accompanying transmission from mosquito vector (28 °C) to human host (37 °C).** Cryo-EM structures of DENV1 and DENV2 icosahedral particles (Triangulation, T = 3) showing symmetry units of E-proteins straddling the five-fold (pentamer), two-fold (dimer) and three-fold (trimer) vertices in dark, medium and light blue respectively. The triangle highlights a single symmetry unit and the numbers indicate the five-fold, three-fold vertices and two-fold midpoint. Left, cryo-EM structure of smooth, compact DENV2 (PDB ID: 3J27, 3.5 Å resolution) and DENV1 (PDB ID: 4CCT, 4.5 Å resolution) at 28 °C (environment inside the mosquito vector). Right, cryo-EM structure of expanded DENV2 (PDB ID: 3ZKO, 13 Å resolution) and unexpanded DENV1 (PDB ID: 4CCT, 4.5 Å resolution) at 37 °C (environment within human host following transmission). White spaces at three-fold icosahedral vertices indicate exposed regions of the lipid bilayer upon expansion. Inset 1: DENV structural proteome. Cryo-EM structures of DENV E (blue) (PDB ID: 3J27) and M (yellow) (PDB ID: 3J27) proteins shown in the context of the viral structure. The lipid bilayer is highlighted in dark grey. Since C-proteins are not observable in the cryo-EM structures of intact DENVs, an approximate position (region coloured red) based on its proposed role in bridging the RNA genome (not shown) beneath the lipid bilayer is represented. Inset 2: NMR structure of C (red) (PDB ID: 1R6R) protein dimer. The N and C-termini of one C-monomer are labelled.

vertices, 20 three-fold faces and 30 two-fold edges. The relative deuterium exchange profiles as well as temperature-dependent differences can be assumed to represent averages of all 180 E-proteins on the virus surface of all viral particles within a sample. The relative exchange across each of the asymmetry units was not different as inferred from the absence of multimodal isotopic envelope profiles or lack of broadening of isotopic envelope profiles for any of the peptides from DENV1 and DENV2. The relative deuterium exchange of the E-protein (Fig. 2a,b,c) mapped onto the cryo-EM structure of the unexpanded DENV2 and DENV1 (Fig. 2d) at 28 °C was used to generate a 'heat-map', which highlighted striking differences between DENV1 and DENV2. Interestingly, deuterium exchange of E-protein on DENV2 was greater compared to the deuterium exchange of E-protein on DENV1 at 28 °C. Peptides spanning glycosylation sites (N67 and N153) on DENV2 E-protein (residues 57-69, 152-163) showed the largest exchange relative to the other fragment peptides (Fig. 2b). In contrast, peptides spanning equivalent glycosylation sites in DENV1 (residues 54-69, 152-173) did not show higher exchange relative to the rest of the protein (Fig. 2c). Similar to E-protein, peptides from C- and M-proteins from both DENV serotypes also showed greater relative deuterium exchange in DENV2 relative to DENV1 at 28 °C (Supplementary Fig. 4a,b).

**DENV2 expansion at 37 °C.** Changes upon temperature-induced expansion (28–37 °C) of DENV2 were monitored by comparing differences in deuterium exchange at 28 and 37 °C for all peptides across the E-, M- and C-proteins. Given that the asymmetry unit arrangements are highly similar across DENV2 at both 28 and 37 °C (Fig. 1), any differences in exchange between the two temperatures would cancel out any ensemble averaging effects. Thus, our readout provides an absolute magnitude difference in temperature-dependent exchange that is independent of asymmetry unit-dependent ensemble averaging.

It should be noted that temperature alters intrinsic rates of amide hydrogen-deuterium exchange, as reported in previous studies[26]. Based on these studies on unfolded model peptides, we estimated the intrinsic rate of HDX to be ∼2.3-fold greater at 37 °C compared to 28 °C (detailed in Supplementary Note 2). The observed increases in deuterium exchange at 37 °C would therefore represent a composite of the intrinsic increases in deuterium exchange rate as a function of temperature together with temperature-dependent conformational changes accompanying DENV2 expansion. We reasoned that for our intact protein and viral particle HDXMS analyses, this 2.3X magnitude increase in intrinsic rates of deuterium exchange would account for a relatively minor contribution to the measured difference in deuterium exchange between DENV at higher and lower temperatures at times of 1 min and longer. This would be expected since the predicted decrease in protection factors in DENV2 with temperature-dependent expansion would be greater relative to this 2.3-fold increase in intrinsic deuterium exchange. To experimentally confirm this, we carried out test studies with HDXMS on post-pepsin hydrolysed unassembled E-protein. In all peptides we were unable to detect observable differences in centroid values between 28 and 37 °C (Supplementary Fig. 5a,b). Further control experiments comparing deuterium exchange measurements on DENV2 labelled for 26 s at 37 °C with that for 1 min at 28 °C established that the estimated 2.3-fold effects of temperature on intrinsic exchange were within the s.e. of deuterium exchange measurements and experimental conditions as well as the 0.5 Dalton significance threshold for defining temperature-specific changes (Supplementary Note 2,

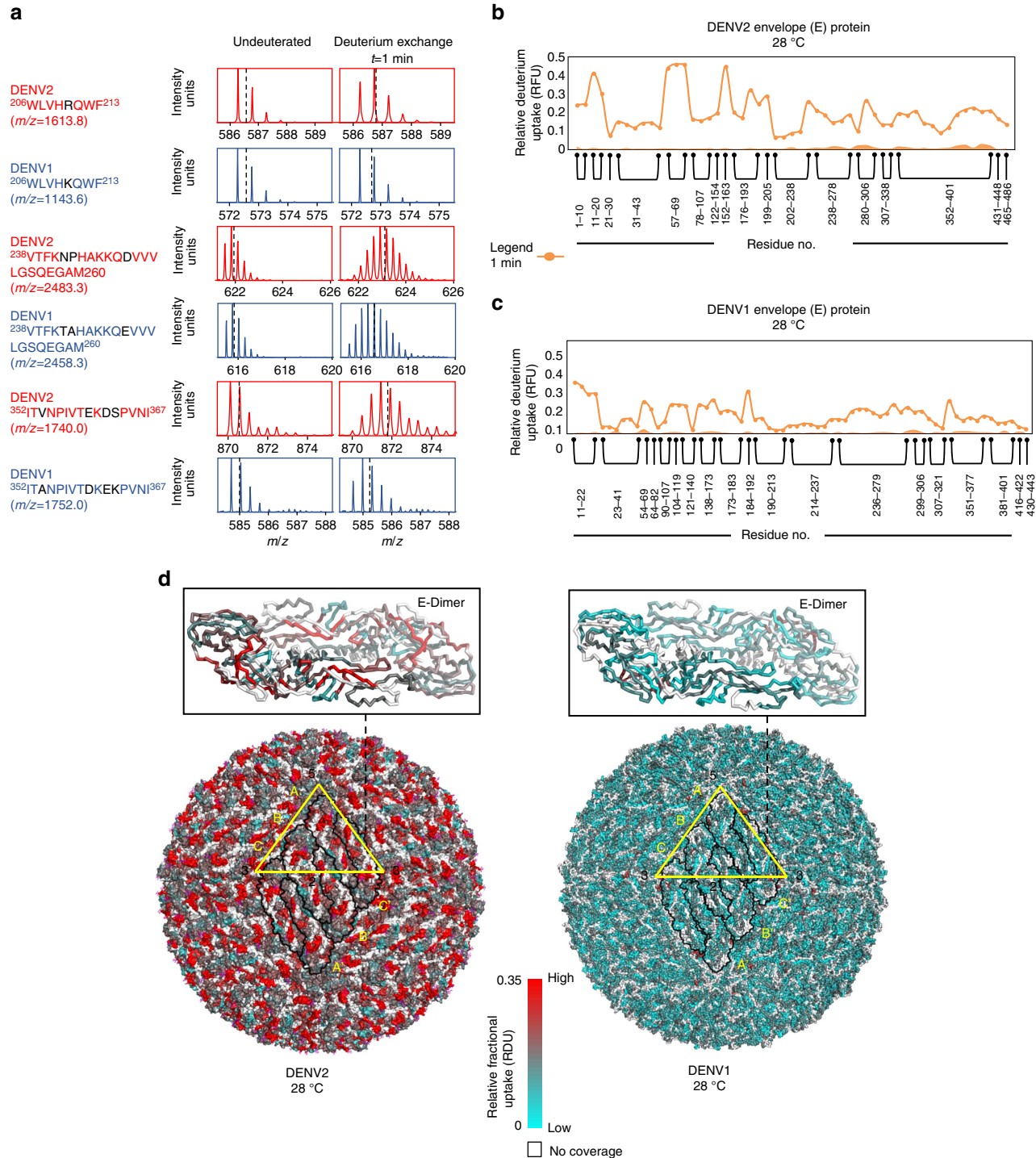

**Figure 2 | Deuterium exchange of E-protein in unexpanded DENV2 and DENV1 at 28 °C in solution.** (**a**) Representative mass spectra of peptides from E-protein of DENV2 and DENV1 for undeuterated and *t* = 1 min of deuterium exchange at 28 °C. Differences in sequence between DENV2 and DENV1 are indicated in black. Dashed line represents centroid of the mass envelope. Average number of deuterons exchanged in each peptide is determined by subtracting the centroid of the deuterium exchanged peptide from the undeuterated peptide. Relative deuterium exchange (RFU) is the fraction of average deuterons exchanged relative to the maximum number of exchangeable amides in each peptide. (**b,c**) RFU values for each E-protein pepsin fragmentation peptide listed from the N-to-C-terminus after 1 min of deuterium exchange are shown for DENV2 and DENV1 at 28 °C in a modified mirror plot respectively. Each dot represents one pepsin fragment peptide, Y-axis-Relative fractional uptake of deuterium (RFU), X-axis-pepsin fragment peptides, listed from N to C terminus. (**d**) RFU after 1 min of deuterium exchange at 28 °C is mapped onto whole viral particles in a colour coded gradient scale on DENV2 (PDB ID: 1OAN) and DENV1 (PDB ID: 4CCT). Regions with no peptide coverage are white. Each E-protein monomer is outlined (consisting of 6 monomers). E-protein units adjacent to the five-fold, two-fold and three-fold vertices are labelled A, B and C respectively. Insets: RFU after 1 min of deuterium exchange at 28 °C mapped onto a unit of E-protein dimer from DENV2 and DENV1. S.e. for each peptide is shown as shaded regions along the X-axis. The s.e. for a given peptide represents a single sigma s.d. of all (minimum three) the independent HDXMS measurements.

Supplementary Figs 5,6a,b, Supplementary Table 1). This confirmed that our measurements under our experimental conditions reported the effects of temperature-dependent expansion of DENV alone.

Temperature-dependent increases in deuterium exchange were observed in specific loci across all three structural proteins of DENV2 and correlated with expansion of the virus structure at 37 °C (Fig. 3a,b and Supplementary Fig. 7). Temperature-specific differences in deuterium exchange within the E-protein (Fig. 3a) align according to the interdimer and intradimer interfaces in the viral surface. Loci showing greater deuterium exchange were mapped onto regions in all ectodomains (I, II, III) of E-protein and included the amphipathic stem helices as well as the transmembrane helices (Fig. 3a,c). Domain II showed the largest temperature-specific differences that were localized to overlapping peptides spanning residues 78-107 and 238-278 (Fig. 3a). The intradimeric interface of E-protein (238-260) in domain II exhibited the greatest difference in deuterons exchanged between 28 and 37 °C. This large increase in deuterons exchanged is indicative of disruption of the interdomain interface upon expansion[13]. Temperature-specific differences were also detected in regions of domain I and III in peptides spanning residues 21-30 and 307-338, respectively (Fig. 3a). Beneath the ectodomains, the second and third stem helices (431-448) and transmembrane helices (465-486) also showed large increases in deuterium exchange at 37 °C (Fig. 3c). Interestingly, the first stem helix and regions (residues 9-17, 19-24 and 283-287) within the E-protein that interact with the first stem helix showed no temperature-dependent differences in deuterium exchange.

We were also able to map temperature-specific differences in exchange in C- and M-peptides unobservable by cryo-EM[13,14]. The N-terminal-regions (1-12), and the stem helix of M-protein (27-45) were observed to show temperature-specific increases in deuterium exchange (Fig. 3b,d) with the stem helix displaying the greatest magnitude difference in deuterium exchange between 28 and 37 °C. Peptides spanning the N-terminal regions of C-protein[27] also showed differences in deuterium exchange between 37 and 40 °C, indicating that the nucleocapsid core of DENV2 also undergoes temperature-specific changes (Supplementary Fig. 7a).

Interestingly, these temperature-specific loci showing concomitant changes on E- and M-proteins align at the E–M interaction interface in the E-M heterotetramer (Fig. 3d). These concomitant increases in deuterium exchange were also observed at the reciprocal interaction surface between the N-terminus of M-protein and the three hydrophobic pockets of E-proteins around the intradimeric interface. These increases in deuterium exchange at both these interfaces are consistent with the observed separation of the E-dimers with expansion as previously reported[13]. Furthermore, the second and third stem helices of E-protein and the stem helix of M-protein also showed temperature-specific increases in deuterium exchange, which indicate a coordinated structural expansion of the E-protein while still maintaining bridging interactions with the lipid bilayer[13]. These observations of E- and M-proteins from HDXMS along with those of the C-protein reveal highly concerted and coordinated responses of the whole intact DENV2 particle to temperature.

No temperature-specific differences were observed across E- and M-peptides in DENV1 at 37 °C compared to 28 °C (Fig. 3e,f). This absence of temperature-dependent differences in deuterium exchange for intact DENV1 particle is in complete contrast to that observed in DENV2 (Fig. 3a,b). Although cryo-EM showed no expansion in DENV1 with temperature, it is significant that the thermodynamic fluctuations of the two strains and serotypes are completely different, with DENV2 showing higher intrinsic fluctuations or dynamics (equilibrium protein motions) relative to DENV1.

**Differential expansion of DENV1 at 40 °C.** After demonstrating that DENV1 virus did not undergo expansion at 37 °C, we set out to test the effects of temperature associated with dengue fever symptoms, 40 °C on DENV1 and DENV2. Surprisingly, our experiments with DENV1 at 40 °C showed significant temperature-specific differences in all three ectodomains of E-protein in DENV1 (Fig. 3e). Peptides spanning residues 307-321, 352-377 in domain III and 152-173 in domain I were observed to display the greatest differences in deuterium exchange between 28 and 40 °C (Fig. 3g). The intradimeric E-interface (238-260) in domain II (Fig. 3e) and the first stem helix (416-422) (Fig. 3g) also exhibited differences in deuterium exchange at 40 °C. Similar to DENV2, these temperature-specific differences were also observed in peptides of the C- (Supplementary Fig. 7b) and M-proteins in DENV1 at 40 °C. An N-terminal peptide of M-protein (1-12) (Fig. 3f,h) that forms the reciprocal interaction interface with the hydrophobic pockets of E-protein also displayed concomitant increases in deuterium exchange at 40 °C. On the other hand, no further temperature-dependent differences in deuterium exchange between 37 and 40 °C were observed across any peptides from E- or M-protein in DENV2.

**Expansion of DENV2 and DENV1 captured by TR-FRET.** Expansion of DENV2 and DENV1 were also captured by TR-FRET to monitor the approximate 15 Å separation of the E-protein from the lipid bilayer based on cryo-EM analysis[13] by monitoring TR-FRET changes between AF488-TFP labelled E-protein (donor) and DiI-C18 labelled lipid bilayer (acceptor) across a temperature range from 25 to 40 °C for DENV2 and DENV1. At 25 °C, the fluorescence lifetime was shorter compared to that at 37 and 40 °C for dual labelled DENV2 (Fig. 4a). These values were also shorter than the negative control with single labelled (donor only) virus particles (($<\tau_{f,DL}>\sim3.3$ ns) and indicated that the labelled E-protein (donor) was closer to the labelled lipid bilayer (acceptor) at 25 °C as compared to that at 37 and 40 °C in DENV2 and DENV1, respectively. By scanning across the temperature range, with a step size of two degrees, from 25 to 37 °C and at 40 °C, we observe a gradual increase in the average lifetimes ($<\tau_{f,DL}>$) from $2.42 \pm 0.05$ ns to $3.18 \pm 0.06$ ns for DENV2 with temperature transitions occurring across 25–37 °C with no further increases from 37 to 40 °C (Fig. 4b). Fluorescence lifetime results are represented as mean ± s.d., from six independent experimental replicates. This is highly consistent with our HDXMS measurements where temperature-specific changes in deuterium uptake were observed only between 28 and 37 °C but not between 37 and 40 °C in DENV2 (Fig. 3a).

DENV1 showed little to no changes in average lifetimes ($<\tau_{f,DL}>$) around $2.37 \pm 0.10$ ns from 25 to 37 °C and increased from $2.37 \pm 0.10$ ns to $3.18 \pm 0.10$ ns when the temperature was raised from 37 to 40 °C (Fig. 4c). This indicated that the temperature transition occurred between 37 and 40 °C in DENV1 and is consistent with our HDXMS results (Fig. 3e). Taken together, the results from HDXMS and TR-FRET revealed that DENV1 expands only at 40 °C whereas DENV2 expansion occurred at 37 °C.

**Temperature-specific loci differ between DENV2 and DENV1.** Even though the respective strains from both DENV serotypes

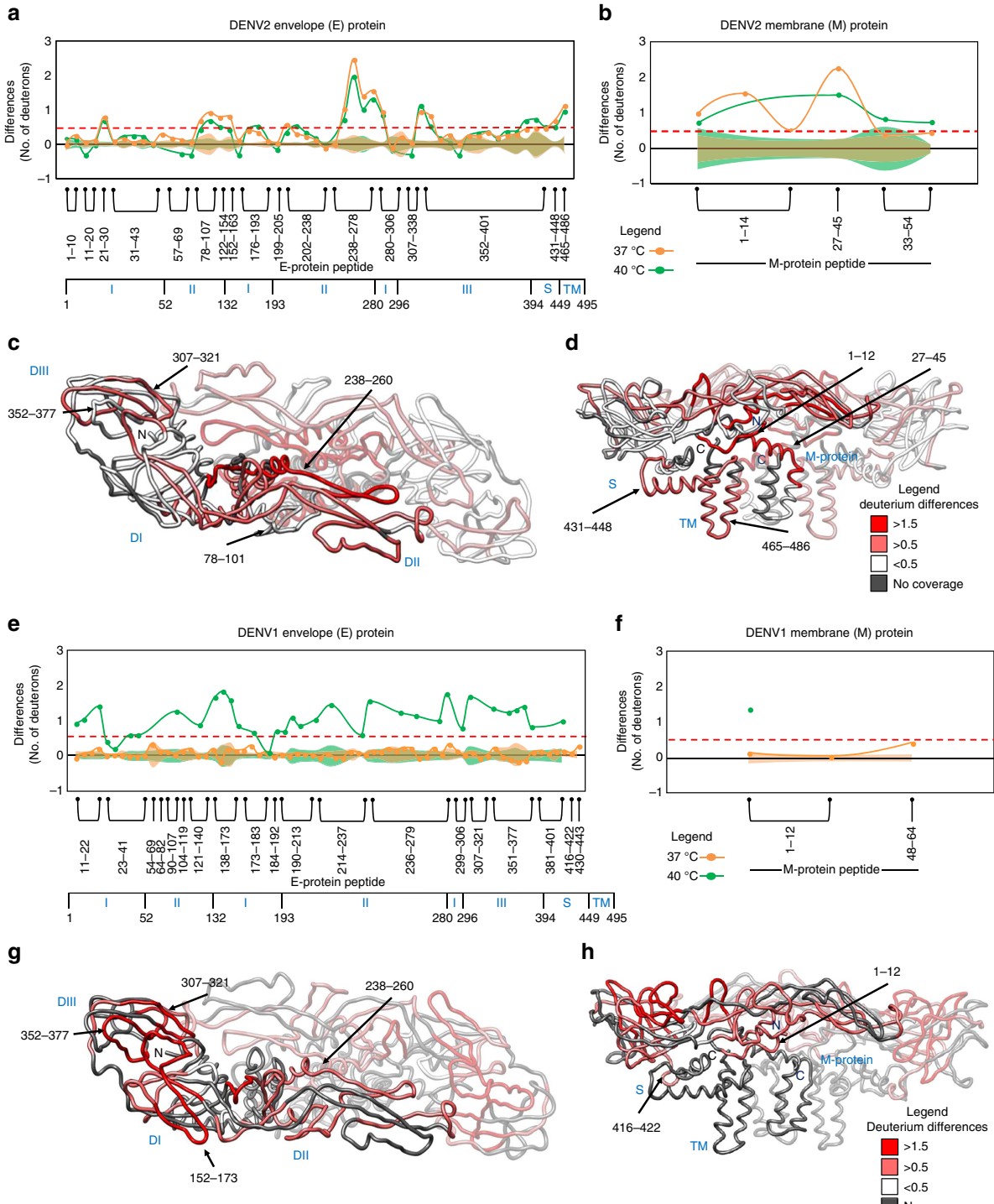

**Figure 3 | Non-uniform temperature-specific changes in DENV2 and DENV1 at 37 and 40 °C by HDXMS.** Temperature-induced differences in deuterium exchange ($t = 1$ min) in E-protein and M-protein from DENV2 (**a**,**b**, respectively) and DENV1 (**e**,**f**, respectively) between 28 and 37 °C (Orange line) and between 28 and 40 °C (Green line) are represented on a difference plot. Difference refers to the difference in average number of deuterons exchanged between two conditions. The difference plot displays the differences in exchange protein-wide where each dot represents a pepsin fragment peptide, listed from the N- to C-terminus. Y-axis-difference in deuterons, X-axis-pepsin fragment peptides. Differences in deuterium exchange above 0.5 D are considered significant (red dash line). Domain organisation of E-protein is indicated below the X-axis. S.e. for each peptide is shown as overlapping shaded regions along the X-axis and coloured according to the conditions in the difference plots. The individual s.e. in each condition is calculated as s.d.'s observed across all the HDXMS measurements (from at least three independent HDXMS experiments). The s.e. for a given peptide in the difference plot represents the sum of such single sigma s.d.'s of each of the two conditions being compared. Orthogonal views of the differences in deuterium exchange in E-protein peptides with temperature (37 °C in DENV2 (**c**,**d**) and 40 °C in DENV1 (**g**,**h**)) mapped onto the cryo-EM structure of E-M heterotetramer with one of the heterodimer rendered transparent for clarity. Regions with no peptide coverage are in grey.

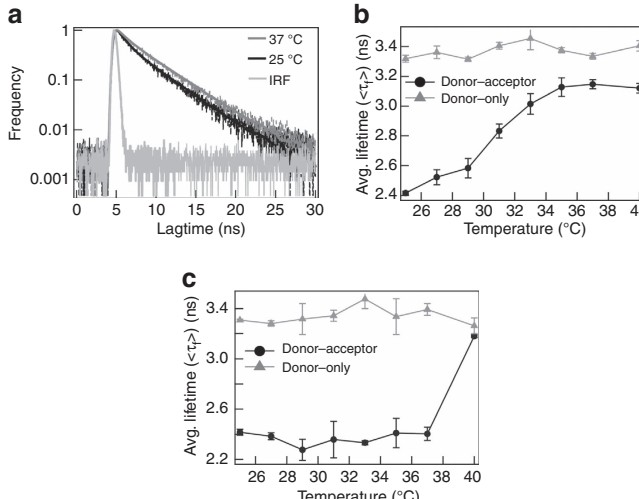

**Figure 4 | Expansion of DENV2 and DENV1 captured by TR-FRET.**
(**a**) Representative fluorescence lifetime decay curve of AF488-TFP for dual labelled DENV2 at 25 (black) and 37 °C (grey), including the impulse response function (IRF) of the instrument. The lifetime traces are fitted with a two-component model. The average fluorescence lifetimes ($<\tau_f>$) of AF488-TFP for donor-only labeled, $<\tau_{f,SL}>$ (grey triangles), and dual labelled, $<\tau_{f,SL}>$ (black circles), for DENV2 (NGC) (**b**) and DENV1 (PVP 159) (**c**). Error bars represent s.d.'s of six different experimental replicates in both DENV2 and DENV1 viruses. The temperature in these two cases was increased from 25 to 40 °C. The midpoint temperature of the transition was at ~33 °C and greater than 37 °C for DENV2 and DENV1, respectively.

undergo expansion and displayed concomitant temperature-dependent differences in HDX at different temperatures (37 or 40 °C), the magnitude differences and loci showing these differences were strain and serotype specific. These differences are clearly evident from comparisons of deuterium exchange in a subset of equivalent pepsin proteolysed fragment peptides of identical length and homology from DENV2 and DENV1. Despite showing high homology, the magnitude of deuterium exchange was different with temperature across the two dengue serotypes (Supplementary Table 2). Peptides spanning the E-intradimeric interface (238-260) showed greater temperature-dependent increases in deuterium exchange in DENV2 as compared to DENV1 (Fig. 5a,b). Whereas, two peptides in domain III (307-321, 352-377) exhibited temperature-dependent increases in deuterium exchange in DENV1 compared to equivalent regions in DENV2 (Fig. 5a,b). Notably, the large temperature-dependent increases in deuterium exchange between 28 and 40 °C in overlapping peptides spanning 352-377 were specific to DENV1 alone (Fig. 5a,b). Mapping these temperature-specific loci onto the respective cryo-EM structures of DENV2 and DENV1 revealed that the largest temperature-dependent increases in deuterium exchange were localized to separate E-protein quaternary contacts along the icosahedral axes between DENV2 and DENV1 (Fig. 5c,d). In DENV2, the E-intradimeric interface (238-260) at the two-fold vertex displayed the greatest temperature-dependent increases in deuterium exchange between 28 and 37/40 °C whereas in DENV1, the E-interdimeric interface (352-377) that lies along the five-fold and three-fold vertices exhibited the greatest temperature-dependent increases in deuterium exchange between 28/37 and 40 °C. This reveals that the thermodynamic hotspots are different between these two respective strains from the two serotypes due to differences in their quaternary contacts.

**Role of viral assembly in DENV structural protein dynamics.**
We further assessed the implications of viral particle-specific quaternary contacts on the equilibrium conformation of proteins in solution by comparing deuterium uptake of unassembled C- and E-proteins from DENV2 and E-protein from DENV1 to their respective proteins in intact DENV particles. A deletion mutant of unassembled E-protein (1-394) that could be stably expressed from both DENV serotypes was purified as described in materials and methods. Unassembled DENV2 C-protein was purchased from commercial vendors as described. Unassembled C- and E-proteins from DENV2 and E-protein from DENV1 were probed for temperature-dependent changes by HDXMS. A map of the pepsin proteolysis fragment peptides of unassembled C- and E-proteins yielded sequence coverage of 73.7, 87.7 and 86.2%, respectively (Supplementary Fig. 8a,b). The relative deuterium uptake values after 1 min of deuterium exchange are mapped onto the structure of the unassembled E-protein from DENV2 and DENV1 (Fig. 6a). This revealed that most regions on the unassembled E-protein from both serotypes showed higher deuterium uptake compared to the intact DENV particles (Figs 2d and 6a–d). A direct comparison of exchange in identical pepsin proteolysed peptides from unassembled E-protein (1-394) and DENV E-protein confirmed this result (Fig. 6e,g).

In DENV2, the intradimeric interface (238-260) and the interface of E-protein that lies along the five-fold and three-fold vertices (352-367) showed the greatest differences in deuterium exchange in unassembled E-protein compared to E-protein on the virus (Fig. 6b,f). Interestingly, both these loci, together with the additional E-interdimeric interface (90-107), also exhibited large differences between unassembled E-protein and E-protein on DENV1. This higher deuterium exchange was also observed in unassembled DENV2 C-protein compared to viral C-protein (Supplementary Figs 4a and 9).

**Importance of quaternary assembly for viral expansion.** The observation that temperature-dependent differences were coordinated across reciprocal interprotein interfaces between E- and M-proteins on intact DENV2 and DENV1 suggests that increased temperature disrupted these quaternary interactions. To test if these temperature-dependent changes were specific to intact viral particles, we examined the effects of temperature on unassembled DENV2 C- and E-proteins and DENV1 E-protein in solution using HDXMS. These individual proteins showed little or no temperature-specific differences (Supplementary Fig. 10a,b). Interestingly, the C-protein from the virion showed temperature-dependent changes in deuterium exchange in both DENV serotypes examined. This suggests that the C-protein is involved in mediating contacts with the rest of the virion. This thus confirms the importance of quaternary structural contacts of E-protein along with other biomolecules such as C- and M-proteins, RNA genome and lipid bilayer in coordinating particle-wide changes in response to temperature and possibly other environmental perturbations.

## Discussion
Viral particles in solution are coordinated dynamic entities that respond to diverse perturbations through their lifecycle, such as temperature, pH, host–protein interactions, among others[12]. HDXMS offers a powerful method to probe dynamics in solution of intact viral particles. By applying HDXMS on two DENV serotypes exhibiting varying temperature-dependent responses, we have captured the equilibrium conformational changes in DENV2 and DENV1 and their constituent proteins in their native state, which encompass protein–protein contacts

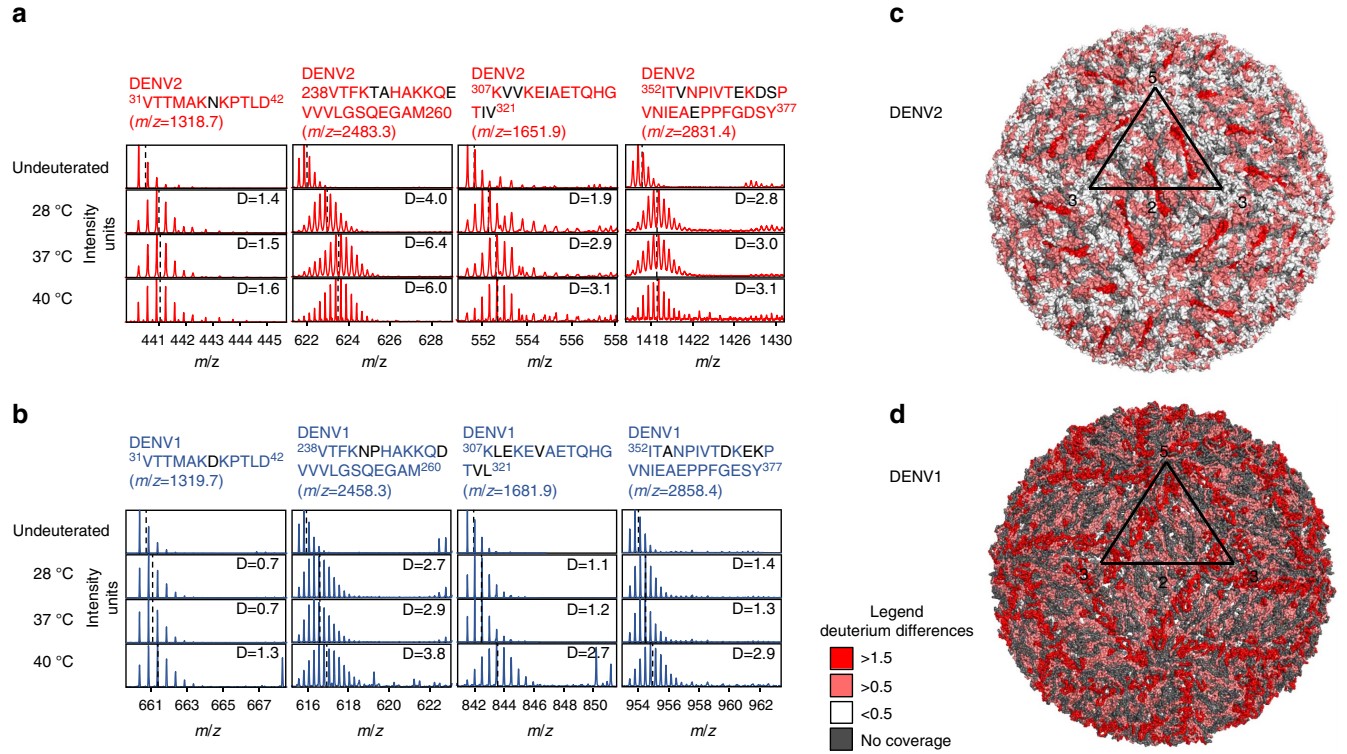

**Figure 5 | Temperature expansion loci differ between DENV2 and DENV1.** Isotopic mass spectral envelopes corresponding to representative homologous peptides of equivalent lengths of DENV2 (**a**) and DENV1 (**b**) E-proteins spanning the two, three and five-fold interaction interfaces after deuterium exchange of $t = 0$ and 1 min at 28, 37 and 40 °C. Dashed lines represent centroid of mass envelope. Differences in sequence between DENV2 and DENV1 are indicated in black. (**c**) Temperature-specific changes at 37 °C mapped onto unexpanded DENV2 structure (PDB ID: 3J27). (**d**) Temperature-specific changes at 40 °C mapped onto unexpanded DENV1 structure (PDB ID: 4CCT). Regions with no peptide coverage are grey.

as well as interactions with genomic RNA and lipids. This has allowed a delineation of temperature-dependent expansion from viral breathing, through parallel analysis of all three DENV structural proteome constituents. This comparative analysis of unassembled and viral E-protein points to important quaternary effects on E-protein packing in modulating dengue viral dynamics and interactions. Strain and serotype-specific expansion of DENV2 and DENV1 were also confirmed with TR-FRET and revealed that DENV2 shows transition in temperature-dependent expansion from 25 to 37 °C with no further changes at 40 °C. DENV1 in contrast showed expansion only at 40 °C.

The RFU mapped onto the E-protein on intact DENV2 shows unsurprisingly, a non-uniform pattern highlighting dynamic 'hotspots' on the viral surface. Interestingly, we observed deuterium exchange to an equivalent extent within the M-protein and to a lesser extent in the C-protein. This highlights the intact virion as a highly dynamic assembly with unobstructed solvent access to M- and C-proteins, despite being invisible from surface maps of whole viral particles. Although the different regions of the E-protein are equally solvent exposed on the surface of the virus, their deuterium uptake values are non-uniform and this is indicative of the importance of H-bonding contributions and quaternary contacts to rates of deuterium uptake. Comparison of the two respective strains from the two serotypes of DENV1 and DENV2 show that despite homologous viral assembly, structure and protein sequences, there are large differences in thermodynamic stability and also in their response to diverse environmental perturbations.

DENV at 37 °C represents the physiologically relevant form of the virus as it enters the human host during infection. At later stages of dengue infection, DENV is further exposed to temperatures as high as 40 °C. These temperature switches enhance viral protein dynamics that drive large rearrangements of the structural proteome and is most apparent in the cryo-EM structure of expanded DENV2 at 37 °C (refs 13,14). Our HDXMS studies revealed that the dimer interface showed the largest temperature-dependent increases in deuterium exchange in DENV2. Careful analysis of the cryo-EM structure of the temperature-expanded virus particle[13] also shows that the dimer interfaces are broken in the B-dimer. On the other hand, the pentamer shows the smallest differences and is the relatively more stable symmetry unit (Fig. 5). The cryo-EM structure also shows minimal differences in this pentamer assembly after expansion (Fig. 1). These temperature-specific loci are in contrast to those observed in DENV1 at 40 °C where the largest temperature-dependent increases in deuterium exchange were observed at interaction interfaces along the three- and five-fold vertices (Fig. 5). These results hold important implications in considering viral particles as integrated thermodynamic units with separate energy thresholds for their expansion governed by a combination of sequence and structural contacts. DENV1 with lower thermal protein fluctuations at 28 °C requires a higher temperature trigger to expand, while DENV2 is more dynamic and expands at 37 °C. Given that the asymmetry unit arrangements are highly similar across DENV2 at both 28 and 37 °C, any contribution of ensemble averaging on differences in exchange between the two temperatures would be cancelled out. Thus, our readout provides an absolute magnitude difference in exchange with temperature that is independent of asymmetry unit-dependent ensemble averaging. This is corroborated by our observation of no

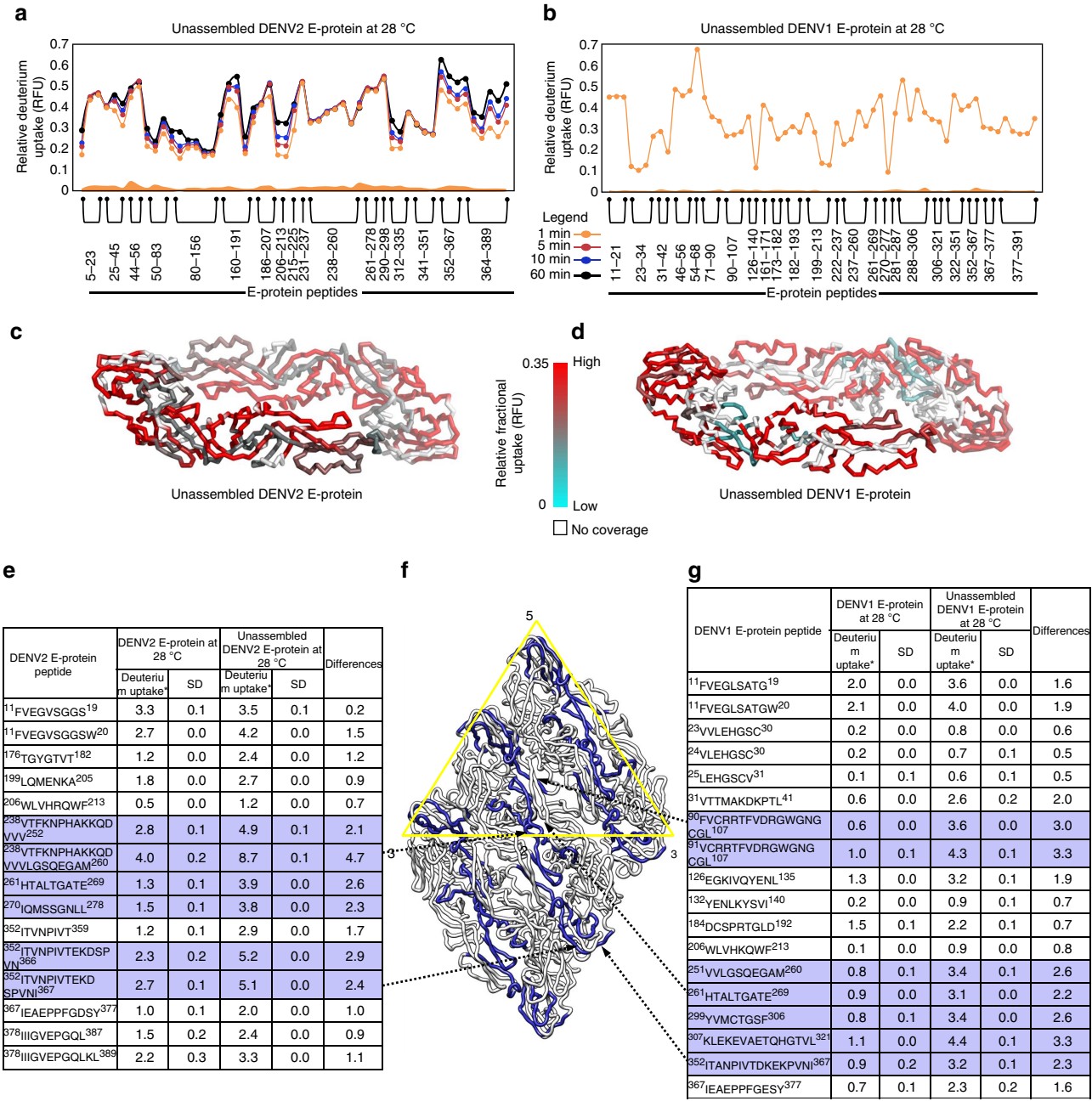

**Figure 6 | Quaternary contacts alter deuterium exchange: Comparison of HDXMS of unassembled E-protein with E-protein from DENV particles.** RFU of unassembled DENV2 E-protein (**a**) at 28 °C for time points (1, 2, 5, 10 and 60 min) and DENV1 E-protein (**b**) at 28 °C for deuterium labelling time point of 1 min are shown as a modified mirror plot. E-protein peptides are listed from N- to C-terminus (X-axis). Error bars for each peptide are shown as shaded regions along the X-axis and are colour coded according to the RFU plots. The s.e. for a given peptide represents the sum of the single sigma s.d.'s over all the time-points from three independent HDXMS measurements. RFU after 1 min of deuterium exchange from unassembled DENV2 (**c**) and DENV1 (**d**) E-protein is overlaid onto the respective E-protein structures (PDB ID:1OAN, 4CCT). RFU is colour coded according to key. The range has been maintained the same as in Fig. 2 for better visual comparison between virion and unassembled E-protein. Comparison of deuterium uptake of identical peptides between viral E-protein and unassembled E-protein from DENV2 (**e**) and DENV1 (**g**). SD represents the single s.d. of deuterium uptake for each listed peptide and differences in deuterium uptake were calculated by subtracting deuterium uptake of unassembled E-protein peptides with viral E-protein peptides at 28 °C. Peptides showing large differences in deuterium exchange (indicated in blue) between unassembled and viral E-proteins in DENV2 and DENV1 are mapped onto a raft of three parallel E-dimers (**f**). Black arrows indicate the regions where the largest magnitude differences in deuterium exchange between unassembled E-protein and viral E-protein were observed.

temperature-dependent effects up to 37 °C across DENV1, which has an identical asymmetry unit distribution as DENV2. Thus the changes in deuterium exchange in DENV2 at 37 °C solely report on conformational changes alone.

It should be noted that structures of DENV1 and DENV2 show all 180 copies of the E-protein in a virion, uniformly arranged into symmetry units (12 pentamers: 20 trimers: 30 dimers), resulting in an equimolar (1:1:1) distribution

of E-protein units into five-fold, three-fold and two-fold vertices, at both 28 and 37 °C temperatures[13] (Fig. 1). The RFU of all peptides from intact DENV2 and DENV1, but not the difference in deuterium exchange with temperature, can be assumed to represent ensemble averages from all three contributing asymmetry units across all DENV peptides. For peptides mediating equivalent quaternary interactions across the icosahedral axes, there would be no signal averaging observed. However, for peptides mediating differential contacts across asymmetry units, signal averaging would need to be factored in absolute deuterium exchange measurements. For instance, the intradimeric interface of E-protein, common across all 90 E-protein dimers in the virion is independent of the icosahedral arrangement, and there would be no contribution of averaging to deuterium exchange measurements. Whereas, an E-protein peptide from DIII in A, B and C arrangements within an asymmetry unit (Fig. 2b,c) would show context-dependent differences in deuterium exchange. This would be attributable to unique contacts mediated by the same peptide in five-fold, five-fold–three-fold interface and the three-fold vertices contexts. Thus the overall deuterium exchange measured for every constituent virion peptide represents a variable average that is a function of the unique quaternary contacts that each peptide possibly mediates in the context of its position within the symmetry unit. If large differences in basal deuterium exchange for a given peptide in each symmetry unit were likely, these would be detected as multimodal isotopic distributions in deuterium exchanged samples, indicative of EX1 kinetics[25,28]. Smaller differences would be evident as extended isotopic envelope distribution patterns due to overlapping exchange of closely exchanging species, indicative of EXX deuterium exchange kinetics[29]. We detected no multimodal isotopic envelopes in any of the virion peptides from DENV1/DENV2, indicating no large differences in exchange in peptides mediating contacts across symmetry units. A closer spectral width analysis of isotopic envelopes in deuterium exchanged peptides from DENV1 and DENV2 and compared to homogenous unassembled E-protein showed no peptides with atypical spectral widths (Supplementary Tables 3 and 4).

HDXMS thus offers a powerful tool for analysis of dynamics in intact viral particles to map temperature-dependent differential expansion in dengue and other flaviviruses. The effects of perturbations would be readily interpretable when there are no perturbation-dependent changes in asymmetry unit distributions, as ensemble averaging effects would cancel out. When representing HDXMS results on viral structures, it is important to take into consideration the unique orientation and surface display of different regions of proteins in each asymmetric unit and any attendant distortion of HDXMS results due to their arrangement. For instance, if a surface is visible in only one out of three asymmetry units in DENV2 and shows large differences in deuterium exchange with temperature, differences in exchange in this peptide would appear to be disproportionately higher than another peptide showing equivalent magnitude difference in deuterium exchange, but hidden from the surface in the two other asymmetry units. Thus, for localizing regions showing the largest magnitude differences without any distortion from the structure itself, it is important to first identify and rank these on the basis of HDXMS difference plots, rather than base it on maps of HDXMS results on a composite structure, such as that of a virus with multiple substructural units.

Among the four serotypes of DENV, the intradimeric interactions of E-protein was predicted to be weakest in DENV2 NGC strain while the interactions between E-proteins along the five-fold and three-fold vertices were predicted to be weakest in DENV1 PVP 159 strain based on cryo-EM[16]. Correspondingly,

we observed at peptide resolution that these loci of weaker interactions displayed the greatest temperature-specific differences in the respective DENV2 (NGC) and DENV1 (PVP 159). Our observations that these temperature-dependent differences are observed only in the whole viral particle but not in the E-protein in solution further emphasized the important contributions of quaternary contacts and interactions with other viral biomolecular constituents. Indeed, the strengths of quaternary interactions have been hypothesized to impact the stability of flaviviruses[30].

Epitope accessibility is one of the primary factors affecting the potency of neutralizing antibodies[31] and the rates and extent of viral structural transitions influences the degree of epitope accessibility, which in turn are dependent upon the underlying dynamics. Many antibodies against flaviviruses exhibit characteristic time and temperature-dependent binding and neutralization[32–34]. Monoclonal antibodies targeting partially or entirely 'cryptic' epitopes on DIII of E-protein such as 1A1D-2, 4E11, 2H12 and E111 (refs 32,35–37) are among the best characterized temperature-dependent antibodies. These antibodies share overlapping epitopes mostly spanning variable regions on DIII A (306-314), B (320-326), D 347-351), E (365-371) and G (387-393) strands with the exception of E111, whose predominant epitopes are localized to strands $C_x$ (332-334) and C (336-341). All peptides spanning each of these characterized epitopes above in both DENV serotypes showed temperature-dependent changes in deuterium exchange (Fig. 5), which is consistent with the increased binding seen at higher temperatures[32,37]. Importantly in both DENV1 and DENV2, the magnitude of deuterium exchange in possible non-homologous, cryptic epitopes differed at lower vector-specific temperatures (28 °C) relative to the human host (37/40 °C). E111, a DENV1-specific mAb, showed a large (~20-fold) enhancement in binding and neutralization at 40 °C relative to 37 °C (ref. 32), which strongly corroborated our findings showing differential expansion of DENV1 at 40 °C.

We further examined the effects of temperature-dependent changes on the E-dimer dependent epitopes (EDE) that are recognized by an important class of broadly neutralizing and highly potent antibodies[38,39] in both DENV1 and DENV2. EDEs are constituted by the $\beta$ strand (67-74), fusion loop (97-106) and ij loop (246-249) in DII from one E-monomer and the 150 loop (148-159) from the adjacent E-protein subunit[38,39]. Interestingly, temperature-dependent increases in deuterium exchange were observed in peptides spanning the fusion loop, ij loop and the 150 loop in both DENV1 and 2. This indicated that the conformation of these epitope sites is likely altered with DENV1 and DENV2 expansion and may influence the binding of these antibodies. Furthermore, in DENV2, the greatest temperature-dependent increase in deuterium exchange was observed at the E-intradimeric interface. This suggests that the EDEs across the E-dimers may be spatially separated in the expanded DENV2. Regardless, EDE antibodies are broadly neutralizing against all DENV serotypes and including the thermally stable Zika virus[40,41]. Therefore, this possibly suggests that these EDE antibodies exert neutralizing effects against all DENV serotypes and Zika virus through multiple mechanisms. Mapping the epitopes of these antibodies on whole DENV particles by HDXMS under various host-specific temperature perturbations will reveal in greater detail the mechanisms of antibody action by this important class of antibodies.

Structures of free viruses do not offer predictive insights into potential antibody binding sites and this represents a limitation of current structural biology tools. Structural studies together with HDXMS measurement of virion dynamics in solution offer a powerful combinatorial approach to identify

potential epitopes, map virus-antibody complex structure and dynamics, and test effects of multiple host-specific perturbations including temperature, pH and osmolality on viruses and virus-antibody complexes.

In summary, HDXMS offers a powerful new probe for probing dynamics of intact virus particles in solution. Our work represents a description of enveloped virion dynamics using whole intact viral particles and mapping temperature-dependent expansion across all three DENV structural proteome constituents in parallel. This is a significant advance for examining individual protein dynamics in the context of the whole proteome with viral genome and lipid components. It provides insights into coordinated motions of viral proteins in the context of host-specific environmental conditions and the ability to differentiate solution properties of homologous viral structures.

## Methods

**Purification of DENV2 NGC and DENV1 PVP159.** DENV2 NGC and DENV1 PVP159 were produced and purified as previously described[9,13]. DENV2 NGC and DENV1 PVP stocks were kind gifts from Michael Rossmann and Eng Eong Ooi, respectively. Briefly, C6/36 *Aedes albopictus* mosquito cells (ATCC CRL-1660) were cultured in Roswell Park Memorial Institute medium (RPMI) medium supplemented with 25 mM HEPES and 10% fetal calf serum (FCS) and grown to ∼80% confluency. The cells were infected with DENV2 NGC/DENV1 PVP159 strain at multiplicity of infection of 0.1 at 28 °C for 2 h. The inoculum was replaced with fresh RPMI medium containing 2% FCS and left to incubate for 4 days. The virus-containing medium was subsequently centrifuged to remove cellular debris and the resulting supernatant was precipitated using 8% polyethylene glycol (PEG 8000) in NTE buffer (12 mM TrisHCl, pH 8.0, 120 mM NaCl and 1 mM EDTA). Virus particles were resuspended in NTE buffer and the amounts of immature DENV particles in the samples were assessed by SDS-PAGE and western blot analysis (Supplementary Fig. 1a). Mature DENV samples were compared with immature DENV samples to confirm the purity of the mature virus preparations. prM protein was identified by a western blot analysis using anti-prM-protein mAb 2H2, diluted to a final concentration of 0.5 ug ml$^{-1}$ in phosphate buffered saline with 0.1% Tween20 and 0.5% bovine serum albumin. Hybridoma producing mAb 2H2 (ATCC HB-114), obtained from American Tissue Culture Collection (ATCC), was grown in BD Cell MAb basal medium (BD Biosciences, Singapore) supplemented with 10% fetal bovine serum. Large-scale antibody production was done in the BD CELLine 1000 culture system (BD Biosciences, Singapore). The mAb was then purified from the tissue culture supernatant using protein-A affinity column (GE Healthcare, Singapore). The secondary antibody used in the western blot analysis was HRP conjugated anti-mouse IgG (H + L) (ThermoFisher (Invitrogen) Cat. No: 62-6520, Singapore) diluted (1:2,000) in PBS with 0.1% Tween20 and 0.5% bovine serum albumin. Since western blot analysis using mAb 2H2 is a highly sensitive method, the prM band in PAGE gel of both the mature and immature virus preparations were confirmed (Supplementary Fig. 1a,b).

The mature DENV samples were subsequently purified through a 24% sucrose cushion and 10–30% potassium tartrate gradient centrifugation. The virus band obtained was extracted, buffer exchanged with NTE buffer and concentrated using Amicon Ultra-4 (100 kDa molecular weight cut-off) to a final volume of approximately 100–150 μl. Concentration of E-protein in the virus sample was estimated by comparison with known bovine serum albumin standards on SDS-PAGE gel stained by Coomassie blue and used as a measure of virus concentration. Concentrations of virus samples used for HDX analysis (0.25 mg ml$^{-1}$) indicated in the following sections thereby correspond to the amount of E-protein in the viral samples. PAGE results showed a very low amount of prM protein (20 kDa) in our mature virus samples compared to an immature virus preparation (Supplementary Fig. 1a). Our SDS-PAGE analysis of mature DENV showed a barely observable faint prM-protein band, whereas the immature DENV samples showed a clear distinct intense prM band (Supplementary Fig. 1a). Our cryo-EM observations are consistent with our PAGE results indicating that a vast majority of particles in our mature virus preparations contain only a few immature particles (Supplementary Fig. 1b). A cryo-electron micrograph of an immature virus sample is provided alongside for reference (Supplementary Fig. 1b). The three distinct bands observed in the SDS-PAGE gel were confirmed to be C-, E- and M-proteins by trypsin digestion and mass spectrometry[42]. Importantly, no tryptic peptides corresponding to pr-region were detected in any of these three distinct bands indicating that the samples contained mainly purified mature DENV particles (Supplementary Fig. 1c).

**Preparation of DENV2 and DENV1 unassembled E-proteins.** *Drosophila melanogaster* Schneider 2 cells (ThermoFisher (Invitrogen Cat. No: R69007, Singapore)) were transfected with pMT/BiP/V5-HisA plasmid encoding soluble deletion mutant of unassembled E-protein (residues 1-394) from DENV2 and DENV1. Unassembled E-protein was glycosylated at the same positions

(N67, N153) as viral E-protein. Unassembled E-protein was purified using 4.8 A affinity column by passing tissue culture supernatant containing unassembled E-protein through the column and washed using 10 mM Tris-HCl and 150 mM NaCl, pH 5 and eluted with 0.1 M Glycine-HCl, pH 2.7. Eluted unassembled E-protein was dialysed with 10 mM Tris-HCl, 150 mM NaCl, pH 7.5 and concentrated to a final concentration of 1.25 mg ml$^{-1}$. Recombinant C-protein from DENV2 NGC strain was purchased from Sino Biological Inc. (Singapore) at a concentration of 0.25 mg ml$^{-1}$.

**Optimal deuterium labelling times for whole DENV2 particles.** Purified DENV2 NGC (0.25 mg ml$^{-1}$) was incubated at 28 °C for 30 min before the hydrogen/deuterium exchange experiments. Deuterium labelling buffer was prepared by resolublizing lyophilized NTE buffer in 99.90% D$_2$O. NTE D$_2$O buffer (12 mM TrisHCl, pH 8.0, 120 mM NaCl, 1 mM EDTA) was similarly incubated at 28 °C to ensure temperature consistency during hydrogen deuterium exchange experiments. HDX was initiated by diluting DENV2 NGC 10X with NTE D$_2$O buffer (99.90%) resulting in a final D$_2$O concentration of 89.91%. DENV2 NGC was subjected to deuterium exchange for 1, 5, 10, 60 min and the temperature was maintained carefully at 28 °C.

**HDX reactions of DENV2 and DENV1 at 28, 37 and 40 °C.** Purified DENV2 NGC (0.25 mg ml$^{-1}$) was incubated at 28, 37 or 40 °C for 30 min and purified DENV1 PVP159 (0.25 mg ml$^{-1}$) was incubated at 28, 37 or 40 °C for 30 min before the hydrogen/deuterium exchange experiments. HDX was initiated by diluting DENV2 NGC and DENV1 PVP159 10X with NTE D$_2$O buffer (99.90%) resulting in a final D$_2$O concentration of 89.91%. DENV2 NGC and DENV1 PVP159 were subjected to deuterium exchange for 1 min and the temperature was maintained carefully at 28, 37 or 40 °C throughout the deuterium exchange reactions as described above.

**HDX reactions of unassembled C- and E-proteins.** Both unassembled DENV2 C- and E-proteins were incubated separately at 28 or 37 °C for 30 min. Hydrogen/deuterium exchange of the two unassembled proteins at 28 and 37 °C were carried out for 1, 5, 10, 60 min as described above. Unassembled DENV1 E-protein was incubated separately at 28, 37 and 40 °C for 30 min and hydrogen/deuterium exchange was carried out for 1 min as described above.

**Assessing temperature effects on intrinsic HDX rates.** In order to assess temperature effects on intrinsic deuterium exchange rates using post-pepsin proteolysed peptides from DENV2 E-protein, purified unassembled DENV2 E-proteins (1-394) were denatured and acidified to pH 2.5 with 1.5 M GnHCl (final) and 2% TFA. Denatured E-proteins were hydrolysed with offline pepsin immobilized on agarose beads (6% slurry) (Pierce, Rockford, IL) for 5 days and pepsin beads were removed with 0.22 μm filter (Merck Millipore, Darmstadt, Germany) and subjected to 1 min of centrifugation at 14,549 g. Deuterium exchange of pepsin proteolysed E-protein peptides was carried out for 3 s and 1 min as described above.

**Quenching HDX reactions.** Deuterium exchange reactions were quenched by lowering the pH$_{read}$ to 2.5 upon addition of pre-chilled NaOH in GnHCl and Tris(2-carboxyethyl) phosphine-hydrochloride (TCEP-HCl) to obtain final concentration of 1.5 M GnHCl and 0.25 M TCEP-HCl. Quenched reactions were maintained at 4 °C on ice to minimize back exchange. Viral membrane phospholipids in the DENV2 NGC and DENV1 PVP159 containing samples were removed by addition of 0.1 mg titanium dioxide (TiO$_2$) (Sigma Aldrich, St. Louis, MO) and incubated on ice for 1 min with mixing every 30 s (ref. 43). TiO$_2$ in the samples were removed with 0.22 μm filter (Merck Millipore, Darmstadt, Germany) and subjected to 1 min of centrifugation at 14,549 g. Lipid removal using TiO$_2$ resulted in an addition of 2 min post-quenching time. In unassembled C- and E-protein, where no phospholipids were present, TiO$_2$ treatment was replaced with 2 min of incubation on ice to ensure equivalent post-quenching sample handling time with both unassembled protein samples and virus samples. All deuterium exchange reactions were performed in triplicate and the reported values for every peptide are an average of three independent reactions without correcting for back exchange.

**Pepsin proteolysis and mass spectrometry analysis.** DENV2 under quenched aqueous conditions was initially tested for optimum pepsin proteolysis under offline and online digestion conditions. Offline pepsin digestion was carried out using pepsin immobilized on agarose beads (6% slurry) (Pierce, Rockford, IL) for 5 min. Quenched samples of DENV2 NGC, DENV1 PVP159, unassembled DENV2 C- (1-100) and E-protein (1-394) and unassembled DENV1 E-protein (1-394) were subjected to online pepsin digestion using Waters enzymate BEH pepsin column (Waters, Milford, MA). Samples (equivalent to ∼100 pmol of E-protein) were injected into a chilled nano-UPLC HDX sample manager (Waters, Milford, MA) as described by Wales *et al.*[44]. Online pepsin digested samples were subjected to pepsin digestion using a 2.1 × 30 mm Waters enzymate BEH pepsin column in 0.05% formic acid in water at 100 μl min$^{-1}$ and trapped

using a 2.1 × 5 mm C18 trap (ACQUITY BEH C18 VanGuard Pre-column, 1.7 µm, Waters, Milford, MA). Peptides were eluted using an 8–40% gradient of acetonitrile in 0.1% formic acid at 40 µl min$^{-1}$ into a reverse phase column (ACQUITY UPLC BEH C18 Column, 1.0 × 100 mm, 1.7 µm, Waters) by nanoACQUITY Binary Solvent Manager (Waters, Milford, MA). Peptides were ionized by electrospray into SYNAPT G2-Si mass spectrometer (Waters, Milford, MA) acquiring in MS$^E$ mode[45] for detection and mass-measurements. 200 fmol µl$^{-1}$ of [Glu$^1$]-fibrinopeptide B ([Glu$^1$]-Fib) was injected at a flow rate of 10 µl min$^{-1}$ into the mass spectrometer for continuous calibration during sample acquisition.

**Pepsin fragment peptide identification.** Sequence identifications were made from mass spectra (MS$^E$) data[45] from undeuterated samples of purified DENV2 NGC and DENV1 PVP159 samples using PROTEIN LYNX GLOBAL SERVER version 3.0 (Waters, Milford, MA) by searching independently against DENV2 NGC and DENV1 PVP159 structural proteome database containing sequences of the C-, E- and M-proteins from the DENV2 NGC and DENV1 PVP159 strain, respectively, with no proteolysis enzyme specified and including variable N-linked glycosylation modifications. The output peptides from each of the undeuterated samples were filtered using a precursor ion mass tolerance of <10 p.p.m. using DynamX (v. 2.0) (Waters, Milford, MA) and with products per amino acids of at least 0.1 and a minimum intensity of 5,000 for both precursor and product ions. Mass spectra from a total of five undeuterated samples were analysed. All but two peptides were independently identified in at least three of five undeuterated samples and met the ion intensity cutoff of 5,000 for precursor and product ions. Two E-protein peptide assignments (206-213, 280-287) observed in two out of five replicates showed a large number of fragmentation ions relative to the peptide size and met the signal intensity cutoff applied. All of the spectra were additionally visually examined and only those with high signal to noise ratios were used for the HDXMS analysis.

Mass spectra (MS$^E$) from post-pepsin proteolysed peptides of DENV2 E-protein and unassembled DENV2 C and E-proteins were used to assign peptides by searches against individual C and E-protein sequences from DENV2 similar to the parameters described earlier with the exception that mass spectra from unassembled C-protein were used to assign peptides by searches against a database of C-protein without proteolysis enzyme specified and no modifications. Mass spectra from DENV1 E-protein were assigned to peptides by searches against individual E-protein sequences from DENV1. Mass spectra from a total of three undeuterated samples were collected and peptides were retained if they were independently identified in a minimum of two out of three undeuterated samples.

We identified four peptides that were glycosylated. These include (57-68, 57-69, 153-162 and 122-154). Fragmentation of these peptides clearly show glycosylation and all of these peptides contain only a single Asn, which has been previously identified as the site of glycosylation[19]. Importantly, glycosylation at known Asn sites was observed in both virions and unassembled E-protein (Supplementary Fig. 11a,b). Peptides that spanned the glycosylation sites were identified using MS$^E$ (ref. 46). Glycans were observed in several of the fragment peptides (Supplementary Fig. 11c).

**Determination of differences in deuterium exchange.** The amount of deuterium uptake for each peptide were determined using DYNAMX Ver. 2.0 software (Waters, Milford, MA) by subtracting the mass centroid of the peptide with the mass centroid of the corresponding undeuterated peptide. The raw deuterium uptake values for every experimental condition (for example DENV2 at 28 °C) are represented in the form of RFU, which normalized to the number of exchangeable amides in the peptide. These RFU values represent normalized deuterium uptake at different loci across the protein primary sequence and are indicators of structural rearrangements or conformational dynamics at these different loci. These are represented in RFU plots and are defined for a single experimental condition.

Differences in deuterium exchange between two different experimental conditions are represented in difference plots. Differences in deuterium exchange for temperature-dependent expansion analysis for all peptides were calculated by subtracting centroid masses of deuterated peptides at 28 °C from their centroid masses at 37 °C. These differences are not normalized and are absolute differences in deuterium uptake across the protein primary sequence between two different temperatures. Deuterium exchange differences were measured for all identified peptides from DENV2 and DENV1 C-, E- and M-proteins and displayed from N to C-terminus. From our experimental conditions and HDXMS setup, we calculated deuterium exchange s.d.'s across all peptides and a difference of 0.5 Da was set as the significance threshold. This also agrees with observed s.e.'s measured in deuterated peptides[47].

**Labelling of DENV2 and DENV1 for TR-FRET.** Alexa Fluor 488 TFP ester (AF488-TFP) and 1,1′-Dioctadecyl-3,3,3′,3′-Tetramethylindocarbocyanine Perchlorate (DiI-C18) were purchased from ThermoFisher Scientific, Singapore. Purified viruses had stock concentrations of 6 × 10$^{10}$ plaque forming units (PFU)/ml for DENV2 (NGC) and 4.2 × 10$^{10}$ PFU/ml for DENV1 (PVP 159). Both DENV1 (PVP 159) and DENV2 (NGC) were dual labelled by sequential addition of DiI-C18 and AF488-TFP, in buffer containing 10 mM HEPES, 150 mM NaCl at pH 7.4 (HN buffer). The DiI-C18 and AF488-TFP stock solutions were prepared in dimethylsulfoxide and subsequently their concentrations were determined using molar extinction coefficients of 144,000 M$^{-1}$cm$^{-1}$ and 71,000 M$^{-1}$cm$^{-1}$, respectively. Initially, the DiI-C18 stock solution was diluted in 50 µl of HN buffer to a final concentration of 100 nM and sonicated for 10 min. Next, ~2.5 × 10$^8$ PFU of the purified viruses were directly added to the DiI-C18 containing HN buffer solution and incubated for 1 h at 4 °C to label the virus-lipid bilayer. Subsequently, AF488-TFP at a final concentration of 750 nM was added to the DiI-C18 labelled virus HN buffer mixture and incubated for an additional hour at room temperature. The free dye molecules were removed by gel filtration (MicroSpinTM S-200 HR columns, GE Healthcare, Singapore). As a control, single labelled donor-only viruses were prepared by labelling with AF488-TFP only.

**TR-FRET measurements.** TR-FRET measurements were carried out on a commercial Olympus FV1200 laser scanning confocal microscope (IX83, Olympus, Singapore) equipped with a time-resolved LSM upgrade kit (Microtime 200, PicoQuant, GmbH, Berlin, Germany). The samples were excited with a 485 nm pulsed diode laser with a 20 MHz repetition rate and 29 µW power (PDL series, Sepia II combiner module). The beam was focused into the sample by a water immersion objective ( × 60, NA 1.2; Olympus, Tokyo, Japan) after being reflected by a dichroic mirror (DM405/485/543/635 band pass, Olympus, Singapore) and the scanning unit. The fluorescence was collected by the same objective followed by a pinhole (120 µm) to remove out-of-focus light. The fluorescence signal was spectrally divided into donor (green) and acceptor (red) channels by a 560DCLP dichroic mirror. The donor fluorescence was recorded by a set of single molecule avalanche photodiodes (SPADs) (SPCM-AQR-14, PerkinElmer Optoelectronics, Quebec, Canada), through a 520/35 band pass emission filter (Omega, VT). This donor signal was further processed by time correlated single photon counting card (TimeHarp 260, PicoQuant) to build up the histogram of photon arrival times.

The temperature of the sample was controlled by an on-stage incubator (TempControl 37-2 digital, Pecon, Erbach, Germany) and an objective heater (TC-124A, Warner Instruments, Hamden, CT). The TR-FRET measurements were recorded for 180 s after incubating single/dual-labelled virus samples for at least 30 min, at a given temperature. The mean lifetime ($\tau$) was calculated from the individual fluorescence lifetimes ($\tau_i$) and their relative amplitudes ($\alpha_i$) according to ($\tau$) = $\Sigma \alpha_i \tau_i$. Donor fluorescence lifetime decay data were treated using the software SymPhoTime 64 (PicoQuant, GmbH). In all cases, the $\chi^2$ values were close to 1 and the weighted residuals as well as their autocorrelations were distributed randomly around 0, indicating good fit. The reported values are mean and s.d.'s from six replicates for both DENV1 (PVP 159) and DENV2 (NGC).

**Data availability.** All data supporting the findings of this study are available within the article and its Supplementary Information files, or are available from the authors upon request.

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

## Acknowledgements

The authors thank Keith Fadgen and Michael Eggertson (Waters Corporation, Milford, MA) for helpful discussions and Dr Miklos Guttman (University of Washington) for modelling estimates of temperature effects on rates of deuterium exchange. Structural mass spectrometry was carried out at the Protein and Proteomics Centre (PPC), Department of Biological Sciences, NUS. The work was supported by Singapore Ministry of Education Tier 3 grant (MOE2012-T3-1-008) awarded to G.S.A., S.-M.L. and T.W. and National Research Foundation Investigatorship award (NRF-NRFI2016-01) the Duke-NUS Signature Research Programme funded by the Ministry of Health, Singapore awarded to S.-M.L.

## Author contributions

A.C. and X.-X.L. equally contributed to this work. A.C., G.S.A. and X.-X.L. designed, analysed and interpreted the results. X.-X.L. performed the HDXMS experiments. A.C., G.S.A. and X.-X.L. wrote the manuscript. N.B., K.K.S. and T.W. performed the TR-FRET experiments and wrote the section on TR-FRET. X.Y.E.L., M.W. and S.-M.L. provided virus samples. A.C., G.S.A., X.-X.L., K.K.S., T.W. and S.-M.L. contributed to manuscript revision.

## Additional information

**Competing financial interests:** The authors declare no competing financial interests.

