## [Peer Review File · Nature Communications]

Reviewers' comments:

Reviewer #1 (Remarks to the Author):

This manuscript describes, for the first time, the use of deuterium exchange techniques for understanding the structure of flaviviruses. The paper is focused on transitions in structure of representative strains of dengue virus 1 and 2 serotypes at several temperatures. The authors claim the use of these methods extends a structural understanding of flaviviruses beyond the limitations of other more static methods. I agree the potential of these approaches is very exciting. The major conclusion of this study is that quaternary contacts are an important determinant of the dynamics of the viral particle. While this is unsurprising given the arrangement of the E proteins on the virion, the details will advance the field and stimulate experimentation.

I am quite enthusiastic about this work and feel a high visible journal is appropriate. That said, I do feel it requires substantial clarification. The importance of the work is that it has the potential to extend beyond the limitations of structural methods that capture only a "snapshot" of a particular structure. What isn't made clear here is the limits of this analysis with respect to the impact of averaging signal among all the E proteins to the analysis (although this is certainly mentioned), quantitative limitations of mass spec, and the heterogeneity of the samples studied (detailed below). The author provide no experimental assurance their conclusions are correct?

Major comments:

* The work on capsid is interesting. A revised manuscript in which this data is deleted (and published elsewhere) might provide the space to provide a more detailed discussion of the author's findings and its interpretation against what is known about flavivirus biology.

* Because all E proteins in the virion are considered equivalent through the lens of deuterium exchange, and the feature in common of E proteins in all three chemical environments is the dimer interface, it is surprising that the majority of the contacts identified for dengue two viruses were at this feature?

* Would the conclusions of the study be the same for viruses lacking N-linked sugars? Does the sugar drive the dynamic states captured here?

* What are the quantitative limitations of the study? What assurance do the authors have that all peptides can be detected equivalently?

* While the Lok/JVI study of the bumpy particle provided nice detail of this elevated temperature form, one very nice feature of this published study is that it also provided a good sense of all the other structures arising from incubation at that elevated temperature. How did the authors account for this in their models? For the presence of subviral particles?

* The authors do not mention uncleaved prM in their analysis. Overwhelming evidence from other studies makes it very unlikely the viruses prepared herein in insect cells are uniformly mature. A Western blot of the preparation to detail heterogeneity is important. How do the authors account for this (and the very different oligomeric state of E when associated with prM).

* Some groups have data suggesting structural transitions involved in antibody binding occur over longer time intervals (>60 minutes), is this possible to capture using these methods? I noted that the time-dependence data provided in the supplement identifies regions where the signal is non-uniform over the four analysis windows.

* How are the authors assured the rapid drop in pH does not result in exchange among surfaces involved in low pH-mediated structural transitions associated with viral fusion?

Minor points:

Clarify first sentence. 390 estimated infections, not clinical cases.

Reference to the intact virion as a viral capsid (and the viral capsid protein) is confusing and not common nomenclature in the field.

How do experiments in references 14-19 measure thermal stability of dengue virus?

Is Zika virus associated with a fever that reaches 40°C? The relevance of this paragraph in the introduction is questionable because the paper is focused on dengue viruses.

E111 appears to be a dengue virus one specific antibody. If dengue one viruses don't assume the bumpy expanded form, how does this transition explain binding? Is there a structure of dengue one bound to this antibody?

The authors should be careful not to generalize with statements about dengue serotype one and dengue serotype two viruses (the original bumpy virus story speaks to the dangers of that). They have studied just two representatives of each serotype and thus cannot/should not, extend their conclusions to all virions in each serogroup (see page 17).

Reviewer #2 (Remarks to the Author):

In the study Lim et al., used hydrogen deuterium exchange mass spectrometry (HDXMS) to analyze temperature dependent conformational changes in two serotypes of dengue virus, DENV1 and DENV2. Previous structural work by electron microscopy demonstrated- at low to modest resolution- that DENV2 undergoes a significant conformational rearrangement when temperature is raised from 28 to 37 °C. DENV1 did not show a similar transition in that temperature range in previous studies. The aim of the present work was to use HDXMS

to provide additional detailed information to show what regions of the DENV proteins and the assembly undergo changes as a function of temperature. They also compare the behavior of both of these serotypes.

I think overall there is very interesting new information in the paper, but in my view, the study would benefit from significant revision or clarification, especially relating to the methodology related to using HDXMS for temperature-dependent analysis.

1) The major weakness of the paper in my view is in the design of the HDX comparison. Temperature changes will offset the intrinsic exchange rates, which will confound the analysis. The authors argue that "This (~ 2.3 -fold) estimated magnitude difference in rates of exchange will not be picked up due to the relatively long timescales of deuterium exchange measurements in our experiments" - but this isn't quite the case. With EX2 kinetics, which one assumes the data reflect (as it is not explicitly stated) the apparent exchange rate will be governed by both the intrinsic exchange rate of the residues in the peptide as well as the protection factor of each region in the context of the native protein.

The authors attempted to use a reference dataset of pre-pepsinized E protein incubated at different temperatures to demonstrate that the temperature dependence of intrinsic exchange of the peptic fragments was negligible, however it is highly likely in this case that all the peptide fragments were fully exchanged already by their earliest 3 second time point. As such, there is no dynamic range with which to actually characterize the temperature dependence or its likely contribution to the intact virus particle HDXMS.

The 2.3 fold exchange rate difference from the temperature could have a significant effect on the observed exchange rate even at longer time points if there is sufficient protection at that region. The easiest option for negating the change in intrinsic exchange rates from temperature differences is to re-scale the time points (~ 20 sec exchange at 37C is equivalent to 1 min at 28C), which is easily done experimentally and will clearly probe differences in structure and conformational dynamics. I'd like to see such a control experiment done to demonstrate that the 37 vs 28°C differences they reported without accounting for temp dependence of exchange gives the identical results.

It does seem that the HDX comparisons are tracking interesting conformational changes triggered by elevated temperatures, but because of the way the study was conducted, it's not readily clear which differences are convincing and which ones may be possible artifacts of the intrinsic exchange offsets. And at present, the quantitative results, i.e. the difference plots, are not yet convincing as a result.

2) Need to validate the structural changes that are taking place by other methods. As such all the data in this study rely upon the single HDXMS technique. Especially the novel transition in DENV1 that takes place somewhere between 37 and 40°C needs to be more rigorously characterized. Seems like a very abrupt change over a narrow range of temperatures.

3) Additional clarification is also needed regarding how peptide assignments were made,

especially glycopeptides, which was not discussed. There are some peptide fragments that are rather unusual considering the general profile of pepsin's cleavage activity. For example DENV2 NGC 57-68 and 122-154 and DENV1 PVP: 54-69 and 138-153. Pepsin tends to not cleave next to glycosylated residues and after prolines (Rapid Commun Mass Spectrom. 2008 Apr;22(7):1041-6). I recommend that the coverage map should be checked and if the glycopeptides are to be included in the paper, there should be some additional way to confirm the peptide assignment. A comparison of the native and deglycosylated sample with PNGaseF after pepsin digestion should provide a good sample for identifying the deglycosylated peptide sequences.

4) Would be worthwhile analyzing spectral widths and testing them for multimodal distributions that may reflect aspects of quasiequivalence. Centroiding alone is not sensitive to these underlying properties and in a sense reduces the information content as well as "dynamic range" or "sensitivity" of the analysis. The authors note this in the Discussion, but a rich body of information that would clarify their assignment of which interfaces open up and which remain more constant is left untapped.

As an example, DENV1 238-260 the mass envelope looks like it's broad and probably bimodal. The authors state that there are multiple oligomeric forms of the E protein, and they only looked at the average, but isn't this throwing out very valuable information? Were there other unexpected bimodals detected? Could more information be gained from trying to quantify - or at least qualitatively examine the different oligomeric species present at the two temperatures?

Minor points:

1) "proteome-wide" is an odd choice of wording as it doesn't typically refer to structural aspects of the proteins. I do get what the authors are trying to convey that they have some (though not complete) information for multiple DENV proteins (E, M, and C). Suggest to find a more readily understood word choice.

2) The language/wording here and there needs to be polished/edited. As just a few examples:

"There are four distinct serotypes of DENV and infection by one serotype does not confer immunity against heterologous serotypes 2 but may lead to antibody-dependent enhancement (ADE)³ leading to development of a more severe life- threatening disease." Sentence structure is odd and "lead" or "leading" is used twice.
"one of possible important triggers"

"these have been visualized in DENV2 as smooth to bumpy surface transition"

Wording: "The high structural homology of DENV1 and DENV2 at 28 {degree sign}C is deceptive, as it does not account for the altered morphological responses to increases in temperature at 37 {degree sign}C." The structures do contain the information that would explain differences in temp sensitivity, however we do not know to interpret the important interactions/residues that might mediate that. Suggest to reword.

3) Coloration and scale of heatmaps and mirror plots don't seem to match in Figures 2 and 4; check all similar figures.

4) Some of the figures lack the lettering for the different panels. (Figure 2& 3)

5) The section in the Introduction about Zika virus seems rather tangential, and could be included in the Discussion instead in a more abbreviated form.

6) Figures 2 and 4; and also Supplementary figures have legends that need editing/proofing. For example the statement "Error bars for each peptide are shown as shaded regions along the X-axis and are color coded according to the RFU plots. Standard error for each peptide is shown as shaded regions along the X-axis." is given including repetitive sentences but also in many of the cases there aren't multiple colors of shading shown; or if one obscures the other colors, it should be stated as such.

7) Page 17: "interraft" meaning of word and use here not clear.

Reviewer #3 (Remarks to the Author):

Mature flavivirus particles principally form smooth spherical assemblies with a herringbone-like icosahedral asymmetric unit. It has been established, however, that the outer protein shell of dengue type 2 virus, which is formed almost exclusively by the envelope protein, E, adopts a more open, so-called bumpy configuration at temperatures of 37°C and above. This configuration exposes a different set of E-protein epitopes. Particles from the other three dengue virus serotypes have not been observed to form bumpy particles by cryoEM. In this study Anand and colleagues probe the accessibility of E protein epitopes in dengue type 1 and type 2 viruses (DENV1 and DENV2) at different physiologically relevant temperatures by amide hydrogen/deuterium exchange mass spectrometry (HDXMS). The data are consistent with DENV2 bumpy particle formation at 37-40°C but the authors find unexpectedly that in DENV1 a distinct set of E protein epitopes is exposed at 40°C (but not at 37°C). The data suggest that DENV1 particles open, or breathe, via a different mode at higher temperatures.

This study demonstrates that DENV1 particles partially open at 40°C following a different breathing mode than DENV2. This unexpected finding has potentially important implications for our understanding of how antibodies may differentially bind and neutralize dengue viruses from different serotypes. Specifically, the distinct set of epitopes exposed to solvent in DENV1 at 40°C may constitute a type 1-specific epitope set that can be targeted by serotype-specific antibodies or used as an antigen in a vaccine. This, and further HDXMS studies may therefore provide a useful complement to serological and structural studies in defining the relevant antibody neutralization epitopes in dengue virus and other flaviviruses. The major concern with the study is that it does not provide a sufficiently strong or detailed link between the HDXMS data and the substantial body of work on dengue virus neutralizing antibody epitope mapping. The HDXMS approach also has the inherent shortcoming that it

provides an average deuterium exchange profile for the E protein and cannot distinguish between the three distinct packing environments of the E protein in the icosahedral assembly of the viral outer shell.

Major concern

1. The major concern with the study is that it does not provide a sufficiently strong or detailed link between the HDXMS data and the substantial available body of work on epitope mapping of various different classes of neutralizing antibodies against dengue virus. In order to achieve its full potential, this study needs to connect the available antibody binding and neutralization data to the new HDXMS data more explicitly and in more detail. As a specific example, do mAbs against epitopes that overlap with residues 352-377 neutralize DENV1 more efficiently than DENV2 at 40°C, as would be predicted by the increased exposure of 352-377 in DENV1 but not DENV2? The authors should discuss this a related connections between the HDXMS and previous epitope mapping data. This should be done without increasing the overall length of the Discussion (see below for suggestions on how to shorten the current text).

Minor points

1. Figure 1 exclusively presents previously published data. While the figure will be helpful to readers less familiar with the topic, it is not essential and could easily be moved to the Supplementary Data.
2. P. 6, paragraph 2. The first half of this paragraph should be cut or shortened.
3. P. 9. The section entitled "Concomitant temperature-dependent differences..." belongs in the Methods or Supplementary Information, not in a standalone section in the Results.
4. Figure 3. The "A" , "B" and "C" panel labels are missing from the figure. This figure has too many panels and subpanels and would be better split into two figures (one with panels A /B and another with C), for example at the expense of moving Fig. 1 to the Suppl. Info.
5. P. 11. "...stem helices... showed temperature-specific increases in deuterium exchange...". This statement appears to be in direct contradiction with the statement on p. 10 "... stem helix and regions that interact with [it] showed no temperature-dependent differences in deuterium exchange". These statements need to be edited so they do not contradict each other.
6. P. 13. "...the icosahedral axis...". Which axis are the authors referring to? Did they mean "axes" (plural)?
7. Figure 4. The "A" and "B" panel labels are missing from the figure. In A iii and iv the worm/sphere radii appear to be different and should be changed so that they are equal.
8. P. 15. The first paragraph of the Discussion is repetitive and redundant. This section should be cut or shortened.
9. P. 18. The last five lines of the first paragraph are repetitive and redundant.

Reviewers' comments:

Reviewer #1 (Remarks to the Author):

This manuscript describes, for the first time, the use of deuterium exchange techniques for understanding the structure of flaviviruses. The paper is focused on transitions in structure of representative strains of dengue virus 1 and 2 serotypes at several temperatures. The authors claim the use of these methods extends a structural understanding of flaviviruses beyond the limitations of other more static methods. I agree the potential of these approaches is very exciting. The major conclusion of this study is that quaternary contacts are an important determinant of the dynamics of the viral particle. While this is unsurprising given the arrangement of the E proteins on the virion, the details will advance the field and stimulate experimentation.

I am quite enthusiastic about this work and feel a high visible journal is appropriate. That said, I do feel it requires substantial clarification. The importance of the work is that it has the potential to extend beyond the limitations of structural methods that capture only a "snapshot" of a particular structure. What isn't made clear here is the limits of this analysis with respect to the impact of averaging signal among all the E proteins to the analysis (although this is certainly mentioned), quantitative limitations of mass spec, and the heterogeneity of the samples studied (detailed below). The author provide no experimental assurance their conclusions are correct?

We thank the reviewer for their insights and suggestions. We agree that HDXMS provides an average ensemble view of dynamics across the different peptides on the virion. However when we map the temperature-dependent changes across DENV1 and DENV2, any contribution of averaging on differences in deuterium exchange between the two temperatures would be cancelled out, since the symmetry unit arrangements are highly similar across DENV2 and DENV1 at all temperatures as seen in the cryo-EM structures. Our HDXMS measurements thus provide a direct readout of the absolute magnitude difference in exchange with temperature that is independent of symmetry unit-dependent ensemble averaging.

Interpretation of absolute deuterium exchange in whole viral particles must factor in averaging and therefore necessitates careful analysis and control experiments and therefore, we have added a new section in the discussion section on pages 18-20 (beginning from the sentence at the last line of p. 18 to the end of the paragraph 1 on p.20), where we have discussed the impact of averaging on interpretation of absolute deuterium exchange in viruses. We have also included a section in supplementary information where we have carried out comparative analysis of spectral width for peptides from isolated E-protein as well as the non-expanding DENV1 to measure any possible impact of signal averaging in our analysis (Supplementary tables 3,4). Here we also describe that we did not observe any multimodal isotopic envelope profiles across any of the peptides in DENV1 and DENV2 in both temperatures. Presence of bimodal/multimodal distributions of isotopic envelopes would reflect broad differences in deuterium exchange in peptides across symmetry units, as a function of possible unique symmetry unit-specific contacts mediated by every peptide.

Further, there was no evidence of broadening of isotopic envelope profiles for any of the peptides which would reflect merger of multimodal isotopic envelopes, attributable to subtler differences in population-specific deuterium exchange. Lastly, we have compared the spectral widths in deuterium exchange profiles for all peptides common between unassembled E-protein and DENV1 /DENV2. These showed no differences (Supplementary tables 3,4). These suggest that the differences in exchange due to symmetry unit-mediated contacts across every peptide are not readily apparent from spectral width and isotopic envelope analysis. This is corroborated by our observation of no temperature-dependent effects upto 37C across DENV1, which has an identical symmetry unit distribution as DENV2. Thus the changes in deuterium exchange in DENV2 at 37 °C solely report on conformational changes alone.

Our HDXMS results represent a bottom-up approach for HDXMS. This approach allows examination of dynamics of large macromolecular assemblies such as viruses in solution by fragmenting the large protein assembly into numerous peptides using the protease pepsin. The range of peptide sizes that we examined were 7 to 25 amino acid residues long. The resolution of our mass spectrometer is ~ 30000- 60000. For small peptides, this offers a high resolution of the isotopic envelopes and a high mass accuracy to measure the number of deuterons exchanged accurately to a high accuracy (<+/- 0.1 Dalton).

All viral samples were purified to high homogeneity. A figure has been included in supplementary information (Supplementary Fig. 1) to show the purity of the sample as seen by polyacrylamide gel electrophoresis. The sample showed little heterogeneity and also shows mature virus particles.

Major comments:

** The work on capsid is interesting. A revised manuscript in which this data is deleted (and published elsewhere) might provide the space to provide a more detailed discussion of the author's findings and its interpretation against what is known about flavivirus biology.*

We thank the reviewer for this suggestion and appreciate their feedback on the significance of our Capsid C-protein results. However, we feel that this paper is strengthened with the results for the membrane M-protein and capsid C-protein also included here. These two proteins, the M and C-proteins are not visible in cryo-EM maps and their contributions are therefore not readily apparent from Cryo-EM. Our analysis with HDXMS is important partly because it is likely the only biophysical tool to probe dynamics in parallel of all constituent proteins in large macromolecular assemblies such as viruses. This also provides important evidence that the structural proteome functions as an integrated coordinated entity when responding to external perturbations such as changes in temperature.

** Because all E proteins in the virion are considered equivalent through the lens of deuterium*

exchange, and the feature in common of E proteins in all three chemical environments is the dimer interface, it is surprising that the majority of the contacts identified for dengue two viruses were at this feature?

In mapping the temperature-dependent changes across DENV1 and DENV2, any contribution of averaging on differences in deuterium exchange between the two temperatures would be cancelled out, since the symmetry unit arrangements are highly similar at the two temperatures. Therefore it is unsurprising that the dimeric interface is the prominent site showing the largest temperature-dependent changes in DENV2 (Fig. 3A). The dimer interface is indeed the common feature in all three symmetry units. Our explanation for the observation that this site is the major locus for temperature-mediated expansion in DENV2 has been included on page 21, paragraph 2. Here we describe how intradimeric interactions of E-protein are predicted to be weakest in DENV2 (from among DENV1, 2 and 4), based on distance measurements in cryo-EM structures and the number of electrostatic interactions and potential H-bonding interactions. Our HDXMS results support this prediction.

Secondly, it is important to take into consideration the unique orientation and surface display of different regions of proteins in each symmetry unit and any attendant distortion when displaying HDXMS results on a surface of the virus which is a composite of the separate symmetry units. For instance, if a surface is visible only in 1 out of 3 symmetry units in DENV2 and shows a large difference in deuterium exchange with temperature, it would appear to be disproportionately prominent on the surface, if other regions that show an equivalent difference in exchange map onto regions that do not show up on the surface in the other two symmetry units. Thus, for inferring the regions showing the largest magnitude differences without any distortion from the structure or its orientation, it is important to first identify and rank these on the basis of difference plots, rather than on the mapped results on a composite structure with multiple substructural units.

** Would the conclusions of the study be the same for viruses lacking N-linked sugars? Does the sugar drive the dynamic states captured here?*

We thank the reviewer for this interesting insight. Glycosylation does confer stability to the proteins and have been shown to greatly reduce the extent of deuterium exchange in other proteins and protein systems. We agree that the glycosylation would have a major effect on the deuterium exchange on DENV. However, our results seem to show that glycosylation does not play a role on temperature-dependent conformational changes as the unassembled E-proteins from both DENV1 and 2, DENV1 and DENV2 (all are glycosylated- Supplementary Fig. 11), show large magnitude differential temperature-dependent changes in deuterium exchange. The effect of N-linked sugars on deuterium exchange of whole viral particles is definitely very interesting and important but is beyond the scope of this manuscript.

** What are the quantitative limitations of the study? What assurance do the authors have that*

all peptides can be detected equivalently?

We have broadly addressed this in our first response. Specifically, our HDXMS results represent a bottom-up approach for HDXMS. This approach allows examination of dynamics of large macromolecular assemblies such as viruses in solution by fragmenting the large protein assembly into numerous peptides using the protease peptides. The range of peptide sizes that we examined were 7 to 25 amino acid residues long. The resolution of our mass spectrometer is ~ 30000- 60000. For small peptides, this offers a very high resolution of the isotopic envelopes and a high mass accuracy to measure the number of deuterons exchanged accurately to a high accuracy (± 0.1 Dalton). This has been added to the methods section in the revised manuscript.

We have used standardized MSE with stringent filters to accurately assign all the peptides from the 3 constituent proteins (C, E, and M) of the structural proteome. These have been described in the methods. Specifically, peptide assignments based on MS^E analyzed by PLGS search software from each of the undeuterated samples were filtered using a precursor ion mass tolerance of < 10 ppm using DynamX (v. 2.0)(Waters, Milford, MA), with products per amino acids of at least 0.2 and a minimum intensity of 5000 for both precursor and product ions. Mass spectra from a total of five undeuterated samples were collected and peptides were retained for HDXMS analysis only if they were independently identified in at least 3 of 5 undeuterated samples. All of the spectra were additionally visually examined and only those with high signal to noise ratio were used for the HDXMS analysis. Since all the peptides analyzed by HDXMS, showed high signal intensities, all the centroid measurements of all peptides showed high accuracy and were equivalent. These are seen in the low standard deviation values for the deuterium exchange seen in both unassembled and virus-assembled states.

** While the Lok/JVI study of the bumpy particle provided nice detail of this elevated temperature form, one very nice feature of this published study is that it also provided a good sense of all the other structures arising from incubation at that elevated temperature. How did the authors account for this in their models? For the presence of subviral particles?*

It is not possible to distinguish deuterium uptake profiles between these viral subpopulations and HDXMS reports on the ensemble average deuterium exchange in solution. There were no evidence for bimodal or multimodal isotopic envelopes in the mass spectra for any of the deuterated peptides. This precludes presence of a mixture of subviral particles with large differences in deuterium exchange between them. Further there was no atypical broadening of spectral widths of mass spectra (Supplementary tables 3, 4). Together these indicate a relatively low level of conformational heterogeneity of viral particles in solution in our experimental conditions.

** The authors to not mention uncleaved prM in their analysis. Overwhelming evidence from other studies makes it very unlikely the viruses prepared herein in insect cells are uniformly mature. A Western blot of the preparation to detail heterogeneity is important. How do the authors account for this (and the very different oligomeric state of E when associated with prM).*

We have found no evidence for uncleaved prM in our viral preparations. prM should migrate as a distinct band larger than the C-protein in polyacrylamide gel electrophoresis. Purified DENV samples contained mainly mature particles as SDS PAGE analysis revealed distinct bands corresponding to the size of C-, E- and M-proteins only. No bands corresponding to prM were observed by SDS PAGE of DENV samples after extensive purification (Supplementary Fig. 1).

** Some groups have data suggesting structural transitions involved in antibody binding occur over longer timer intervals (>60 minutes), is this possible to capture using these methods? I noted that the time-dependence data provided in the supplement identifies regions where the signal is non-uniform over the four analysis windows.*

We thank the reviewer for this interesting point. HDXMS is suitable to examine structural and conformational changes occurring at slower time scales ($t > 60$ min). This can be examined by preincubation of saturating concentrations of antibody with virion and then carrying out deuterium exchange over long time scales.

** How are the authors assured the rapid drop in pH does not result in exchange among surfaces involved in low pH-mediated structural transitions associated with viral fusion?*

We do not believe that there would be any low pH-mediated structural transitions such as associated with viral fusion for the following reasons: 1) HDXMS captures exchange in the protein or virion at the point of time just preceding the point at which the pH is rapidly dropped to pH 2.5. This quenches the deuterium exchange reaction and greatly slows down the rate of exchange (6-7 orders of magnitude slower), allowing for measurement of deuterons exchanged by mass spectrometry. The rapid drop in pH to 2.5 would not capture any low pH-mediated structural transitions which are observed at pH 5. 2) Importantly, the drop in pH to 2.5 is accompanied by simultaneous addition of large concentrations of denaturant (Guanidinium HCl at a final concentration of 1.75M) and reducing agent. This would greatly destabilize the virion and obviate formation of any stable pH-induced structural transitions in dengue virus.

Minor points:

Clarify first sentence. 390 estimated infections, not clinical cases.

This has been done.

Reference to the intact virion as a viral capsid (and the viral capsid protein) is confusing and not common nomenclature in the field.

We have replaced capsid proteome with structural proteome of the virion throughout the manuscript, including the title.

How do experiments in references 14-19 measure thermal stability of dengue virus?

We thank the reviewer for pointing this out. We note that the threshold temperature for virus expansion across different dengue strains need not necessarily correlate with the thermal stability of the virus. We have entirely removed this point from the text.

Is Zika virus associated with a fever that reaches 40C? The relevance of this paragraph in the introduction is questionable because the paper is focused on dengue viruses.

We agree with the reviewer. This paper's focus is on dengue virus and not zika virus. We have removed all references to Zika virus from the text.

E111 appears to be a dengue virus one specific antibody. If dengue one viruses don't assume the bumpy expanded form, how does this transition explain binding? Is there a structure of dengue one bound to this antibody?

We have added an extensive section in the discussion on page 21 (last section, 3 lines from the bottom) to highlight the relevance of our work to reported epitope identification and antibody interactions. It has been seen that E111, a DENV1 specific mAb, showed an ~ 20-fold enhancement in binding and neutralization at 40 °C relative to 37 °C (reference 35). This strongly corroborated our findings which highlighted the unique expansion of DENV1 at 40 °C relative to 37 and 28 °C. This 40 °C-specific distinct expansion in DENV1 likely facilitates improved binding of E111.

The authors should be careful not to generalize with statements about dengue serotype one and dengue serotype two viruses (the original bumpy virus story speaks to the dangers of that). They have studied just two representatives of each serotype and thus cannot/should not, extend their conclusions to all virions in each serogroup (see page 17).

We thank the reviewer for highlighting this. We have strived to emphasize that our results represent a representative strain of each serotype and not all serotypes. We have amended this section and throughout the entire manuscript by including the strain name in all references to DENV1 and DENV2.

Reviewer #2 (Remarks to the Author):

In the study Lim et al., used hydrogen deuterium exchange mass spectrometry (HDXMS) to analyze

temperature dependent conformational changes in two serotypes of dengue virus, DENV1 and DENV2. Previous structural work by electron microscopy demonstrated- at low to modest resolution- that DENV2 undergoes a significant conformational rearrangement when temperature is raised from 28 to 37 °C. DENV1 did not show a similar transition in that temperature range in previous studies. The aim of the present work was to use HDXMS to provide additional detailed information to show what regions of the DENV proteins and the assembly undergo changes as a function of temperature. They also compare the behavior of both of these serotypes.

I think overall there is very interesting new information in the paper, but in my view, the study would benefit from significant revision or clarification, especially relating to the methodology related to using HDXMS for temperature-dependent analysis.

1) The major weakness of the paper in my view is in the design of the HDX comparison. Temperature changes will offset the intrinsic exchange rates, which will confound the analysis. The authors argue that "This (~2.3-fold) estimated magnitude difference in rates of exchange will not be picked up due to the relatively long timescales of deuterium exchange measurements in our experiments" - but this isn't quite the case. With EX2 kinetics, which one assumes the data reflect (as it is not explicitly stated) the apparent exchange rate will be governed by both the intrinsic exchange rate of the residues in the peptide as well as the protection factor of each region in the context of the native protein.

The authors attempted to use a reference dataset of pre-pepsinized E protein incubated at different temperatures to demonstrate that the temperature dependence of intrinsic exchange of the peptic fragments was negligible, however it is highly likely in this case that all the peptide fragment were fully exchanged already by their earliest 3 second time point. As such, there is no dynamic range with which to actually characterize the temperature dependence or its likely contribution to the intact virus particle HDXMS.

The 2.3 fold exchange rate difference from the temperature could have a significant effect on the observed exchange rate even at longer time points if there is sufficient protection at that region. The easiest option for negating the change in intrinsic exchange rates from temperature differences is to re-scale the time points (~20 sec exchange at 37C is equivalent to 1 min at 28C), which is easily done experimentally and will clearly probe differences in structure and conformational dynamics. I'd like to see such a control experiment done to demonstrate that the 37 vs 28°C differences they reported without accounting for temp dependence of exchange gives the identical results.

It does seem that the HDX comparisons are tracking interesting conformational changes triggered by elevated temperatures, but because of the way the study was conducted, it's not readily clear which differences are convincing and which ones may be possible artifacts of the intrinsic exchange offsets. And at present, the quantitative results, i.e. the difference plots, are not yet convincing as a result.

We thank the reviewer for their feedback and suggestions. We have accepted their suggestion to compare deuterium exchange in DENV2 at 28 °C for 1 min with that at 37 °C for 26 s (predicted 2.3X faster intrinsic deuterium exchange due to the shift from 28 °C to 37 °C). We found no significant differences in deuterium exchange for all the peptides. These are

included in Supplementary figure 6 which shows the difference plot of deuterium exchange at 37 °C for 26 s with that at 28 °C for 1 min. The peptides that show temperature-dependent changes in deuterium exchange are highlighted in orange in supplementary table 1. A comparison of the deuterium exchange in all other peptides showed no significant differences in exchange (most peptides showed a difference within ± 0.2 , with one peptide alone showing a difference of 0.4). The range of differences in absolute deuteration (± 0.2) fell within the standard deviation from triplicate measurements falling within ± 0.2 (only 2 out of 40 peptides showed a standard deviation of 0.4). We conclude thus that the deuterium exchange measured in all virion peptides was equivalent and solely reporting on temperature effects on virion conformation. Further, our significance threshold of 0.5 Deuterons is sufficient to filter for experimental variations and temperature effects on intrinsic deuterium exchange. Any peptides showing > 0.5 D differences in deuterium exchange between the two temperatures, reports entirely on the global effects of temperature on conformation.

2) Need to validate the structural changes that are taking place by other methods. As such all the data in this study rely upon the single HDXMS technique. Especially the novel transition in DENV1 that takes place somewhere between 37 and 40°C needs to be more rigorously characterized. Seems like a very abrupt change over a narrow range of temperatures.

We have included a section on time resolved fluorescence FRET where we have shown using tr-FRET that DENV1 does indeed undergo a sharp expansion between 38 and 40C. Comparatively, in DENV2, the expansion is more gradual and correlate well with our HDXMS. We have included these results (new Fig. 4) and this provides additional supporting evidence and detail for the temperature-dependent expansion in both serotypes and specifically at 40 °C in DENV1.

3) Additional clarification is also needed regarding how peptide assignments were made, especially glycopeptides, which was not discussed. There are some peptide fragments that are rather unusual considering the general profile of pepsin's cleavage activity. For example DENV2 NGC 57-68 and 122-154 and DENV1 PVP: 54-69 and 138-153. Pepsin tends to not cleave next to glycosylated residues and after prolines (Rapid Commun Mass Spectrom. 2008 Apr;22(7):1041-6). I recommend that the coverage map should be checked and if the glycopeptides are to be included in the paper, there should be some additional way to confirm the peptide assignment. A comparison of the native and deglycosyled sample with PNGaseF after pepsin digestion should provide a good sample for identifying the de-glycosylated peptide sequences.

We have included a section in methods for describing how we assigned peptides spanning the two known glycosylation sites, N67 and N153. Additionally, we have also described in greater detail the parameters for peptide modification and protease specificity. We included N-linked glycosylation modifications as allowed modifications in searches for all peptides from unassembled and viral E-proteins from DENV1 and DENV2.

We have rechecked our pepsin digestion peptides for all constituent proteins (C, E and M) across both DENV1 and DENV2. All were consistent with the specificity of pepsin cleavage sites reported by Hamuro et al. (2008)- (Rapid Commun Mass Spectrom. 2008 Apr;22(7):1041-6). In the Hamuro et al. study, glycosylated proteins were not included in the analysis and thus it is not known how pepsin cleaved glycosylated peptides and what its cleavage preferences were for glycosylated residues.

None of the E-protein pepsin fragment peptides in our study indicated any pepsin cleavage after either the two glycosylation sites. Our glycosylated pepsin fragment peptide identification showed one peptide cleavage 1 or 2 residues C-terminal to the glycosylation site (peptides 57-68, 57-69). These are consistent with another study that we have now cited (ref #45 Xie H, *et al.* Rapid comparison of a candidate biosimilar to an innovator monoclonal antibody with advanced liquid chromatography and mass spectrometry technologies. *MAbs* 2, 379-394 (2010).in revised manuscript) on pepsin cleavage of glycosylated proteins which showed that pepsin cleavage post 1 residue after the glycosylation site was not disallowed. We have included additional description in the methods section for how glycosylation was considered in peptide identification in E-protein.

We have included a figure in supplementary information (Supplementary Fig. 11) to show glycan fragmentation of identified glycopeptides. This confirms accurate identification of the glycosylated peptide by MS^E and that all E-proteins analyzed (unassembled and assembled E-proteins from DENV1 and DENV2) showed N-linked glycosylation at the reported Asn sites (Asn 67 and Asn 153). We therefore respectfully disagree with the reviewer that PNGaseF after pepsin digestion for identifying the de-glycosylated peptide sequences would not be necessary in our experiment.

4) Would be worthwhile analyzing spectral widths and testing them for multimodal distributions that may reflect aspects of quasiequivalence. Centroiding alone is not sensitive to these underlying properties and in a sense reduces the information content as well as "dynamic range" or "sensitivity" of the analysis. The authors note this in the Discussion, but a rich body of information that would clarify their assignment of which interfaces open up and which remain more constant is left untapped.

As an example, DENV1 238-260 the mass envelope looks like it's broad and probably bimodal. The authors state that there are multiple oligomeric forms of the E protein, and they only looked at the average, but isn't this throwing out very valuable information? Were there other unexpected bimodals detected? Could more information be gained from trying to quantify - or at least qualitatively examine the different oligomeric species present at the two temperatures?

We carried out a careful examination of all the mass spectral data and have included spectral width analysis of all peptides and these are reported in Supplementary tables 3,4. We did not observe any differences in spectral width between deuterium exchanged mass spectral envelope distributions of peptides from DENV1/2 relative to homogenous unassembled E-protein. We agree with the reviewer that a very slight broadening in mass spectral envelope for

238-260 is observed in DENV1 and 2, but not significantly different from unassembled E-protein from DENV2. Larger peptides such as this particular one with a length of 22 amino acids are likely to show a propensity for assuming partial secondary structural arrangements and this is the likely explanation for the slight broadening seen in both unassembled and viral E-protein. We therefore do not believe this slight broadening in spectral width reflects differential exchange across symmetry units. The spectral width comparison for this peptide from DENV2 is shown in supplementary table 3.

We agree that centroids represent only one read out of HDXMS analysis. However, we are unable to conclude on other aspects as we did not detect any atypical spectra. We have included a section in the discussion, where we state that we did not observe any multimodal/ bimodal spectral envelopes or spectral broadening and this showed that there are no large differences in deuterium exchange between the substructural units/ or the three symmetry units in the virion. This is a new section that we have added to the discussion section on pages 18-20 which in addition addresses the point of signal averaging raised by reviewer 1.

Minor points:

1) *"proteome-wide" is an odd choice of wording as it doesn't typically refer to structural aspects of the proteins. I do get what the authors are trying to convey that they have some (though not complete) information for multiple DENV proteins (E, M, and C). Suggest to find a more readily understood word choice.*

One of the strengths of HDXMS for probing whole viral particle dynamics is that it allows simultaneous mapping of changes across all the constituent proteins or the structural proteome. We therefore feel that this word best conveys the results of our study. However, we accept the reviewer's suggestion and have amended the title to "structural proteome". Further, in the abstract and at the first instance in the introduction, where we describe proteome-wide changes, we define proteome-wide to encompass the three structural proteins C, E and M.

2) *The language/wording here and there needs to be polished/edited. As just a few examples:
"There are four distinct serotypes of DENV and infection by one serotype does not confer immunity against heterologous serotypes 2 but may lead to antibody-dependent enhancement (ADE)3 leading to development of a more severe life- threatening disease." Sentence structure is odd and "lead" or "leading" is used twice.
"one of possible important triggers"*

"these have been visualized in DENV2 as smooth to bumpy surface transition"

Wording: "The high structural homology of DENV1 and DENV2 at 28 {degree sign}C is deceptive, as it does not account for the altered morphological responses to increases in temperature at 37 {degree sign}C." The structures do contain the information that would explain differences in temp sensitivity,

however we do not know to interpret the important interactions/residues that might mediate that. Suggest to reword.

We thank the reviewer for their careful examination of the text. We have amended all the sentences above as suggested and have also carefully checked the entire manuscript again for any other such instances.

3) Coloration and scale of heatmaps and mirror plots don't seem to match in Figures 2 and 4; check all similar figures.

The heat map scale based on the relative deuterium exchange was intentionally kept the same, in order to compare the relative exchange in the unassembled E-protein with that assembled on the virion in the same scale for easier visual comparison. We have defined the deuteration heat map range in figure legend 4.

4) Some of the figures lack the lettering for the different panels. (Figure 2& 3)

We have included lettering for all panels.

5) The section in the Introduction about Zika virus seems rather tangential, and could be included in the Discussion instead in a more abbreviated form.

We agree and have removed the section and all references to Zika virus from the manuscript.

6) Figures 2 and 4; and also Supplementary figures have legends that need editing/proofing. For example the statement "Error bars for each peptide are shown as shaded regions along the X-axis and are color coded according to the RFU plots. Standard error for each peptide is shown as shaded regions along the X-axis." is given including repetitive sentences but also in many of the cases there aren't multiple colors of shading shown; or if one obscures the other colors, it should be stated as such.

We have removed the repetition in sentences in figure legends.

7) Page 17: "interraft" meaning of word and use here not clear.

We have replaced the term interraft with a more detailed description- "the intradimeric interactions of E- protein was predicted to be weakest in serotype 2 NGC strain while the interactions between E-proteins along the five-fold and three-fold vertices" in p.21 in the revised manuscript.

Reviewer #3 (Remarks to the Author):

Mature flavivirus particles principally form smooth spherical assemblies with a herringbone-like icosahedral asymmetric unit. It has been established, however, that the outer protein shell of dengue type 2 virus, which is formed almost exclusively by the envelope protein, E, adopts a more open, so-called bumpy configuration at temperatures of 37°C and above. This configuration exposes a different set of E-protein epitopes. Particles from the other three dengue virus serotypes have not been observed to form bumpy particles by cryoEM. In this study Anand and colleagues probe the accessibility of E protein

epitopes in dengue type 1 and type 2 viruses (DENV1 and DENV2) at different physiologically relevant temperatures by amide hydrogen/deuterium exchange mass spectrometry (HDXMS). The data are consistent with DENV2 bumpy particle formation at 37-40°C but the authors find unexpectedly that in DENV1 a distinct set of E protein epitopes is exposed at 40°C (but not at 37°C). The data suggest that DENV1 particles open, or breathe, via a different mode at higher temperatures.

This study demonstrates that DENV1 particles partially open at 40°C following a different breathing mode than DENV2. This unexpected finding has potentially important implications for our understanding of how antibodies may differentially bind and neutralize dengue viruses from different serotypes. Specifically, the distinct set of epitopes exposed to solvent in DENV1 at 40°C may constitute a type 1-specific epitope set that can be targeted by serotype-specific antibodies or used as an antigen in a vaccine. This, and further HDXMS studies may therefore provide a useful complement to serological and structural studies in defining the relevant antibody neutralization epitopes in dengue virus and other flaviviruses. The major concern with the study is that it does not provide a sufficiently strong or detailed link between the HDXMS data and the substantial body of work on dengue virus neutralizing antibody epitope mapping. The HDXMS approach also has the inherent shortcoming that it provides an average deuterium exchange profile for the E protein and cannot distinguish between the three distinct packing environments of the E protein in the icosahedral assembly of the viral outer shell.

Major concern

1. The major concern with the study is that it does not provide a sufficiently strong or detailed link between the HDXMS data and the substantial available body of work on epitope mapping of various different classes of neutralizing antibodies against dengue virus. In order to achieve its full potential, this study needs to connect the available antibody binding and neutralization data to the new HDXMS data more explicitly and in more detail. As a specific example, do mAbs against epitopes that overlap with residues 352-377 neutralize DENV1 more efficiently than DENV2 at 40°C, as would be predicted by the increased exposure of 352-377 in DENV1 but not DENV2? The authors should discuss this a related connections between the HDXMS and previous epitope mapping data. This should be done without increasing the overall length of the Discussion (see below for suggestions on how to shorten the current text).

We thank the reviewer for their insight and suggestion to discuss the correlation of our biophysical measurements with known antibody binding sites and broader implications on flavivirus biology. We have included a section in the discussion (p.22) describing all the structural characterization of antibody interactions on dengue virus previously reported, particularly those that are sensitive to temperature.

We have described the implications of signal averaging and specify that these would not impact our map of temperature-dependent changes in DENV2 in our response to reviewer 1.

Minor points

1. Figure 1 exclusively presents previously published data. While the figure will be helpful to readers less familiar with the topic, it is not essential and could easily be moved to the Supplementary Data.

We acknowledge the reviewer's point that Figure 1 entirely represents previously published data. However, we do prefer to retain this figure in the main text. In addition to highlighting to less familiar readers, the background and significance of the study, we wish to show the highly similar symmetry unit distribution between DENV2 at 28 °C and 37 °C, despite the significant expansion observed at the higher temperature. This would offer readers a visual aid when we discuss the effects of signal averaging as a function of the unique E-protein contacts in the three symmetry units.

2. P. 6, paragraph 2. The first half of this paragraph should be cut or shortened.

We have condensed the description of HDXMS and introduced a paragraph describing time-resolved FRET- as a new biophysical technique for tracking temperature-mediated transitions in DENV expansion as suggested by reviewer 2.

3. P. 9. The section entitled "Concomitant temperature-dependent differences..." belongs in the Methods or Supplementary Information, not in a standalone section in the Results.

We agree. We have moved this to supplementary information.

4. Figure 3. The "A", "B" and "C" panel labels are missing from the figure. This figure has too many panels and subpanels and would be better split into two figures (one with panels A /B and another with C), for example at the expense of moving Fig. 1 to the Suppl. Info.

We have added necessary labels.

5. P. 11. "...stem helices... showed temperature-specific increases in deuterium exchange...". This statement appears to be in direct contradiction with the statement on p. 10 "... stem helix and regions that interact with [it] showed no temperature-dependent differences in deuterium exchange". These statements need to be edited so they do not contradict each other.

We thank the reviewer for pointing this out. We have clarified the text to state that it is specifically second and third stem helices alone that show temperature-dependent changes on p.11.

6. P. 13. "...the icosahedral axis...". Which axis are the authors referring to? Did they mean "axes" (plural)?

Yes. We thank the reviewer for pointing this out. We have changed the text on pages 15 and 19.

7. Figure 4. The "A" and "B" panel labels are missing from the figure. In A iii and iv the worm/sphere radii appear to be different and should be changed so that they are equal.

We have switched the representation from sphere to ribbon and this now appears uniform.

8. P. 15. The first paragraph of the Discussion is repetitive and redundant. This section should be cut or shortened.

9. P. 18. The last five lines of the first paragraph are repetitive and redundant.

We have addressed the redundancies raised in points 8 and 9 and reworded the discussion.

Thank you.

Sincerely,

Ganesh S. Anand

Reviewers' comments:

Reviewer #1 (Remarks to the Author):

*** The PAGE gel of purified virus is a nice approach to demonstrate the relative cleavage of prM in their preparations, provided the authors are confident they have correctly assigned each viral gene product to each band. Evidence to support this is lacking. The conclusion that DENV2 NGC virus produced by culture in C636 insect cells is efficiently cleaved by furin would be unprecedented (and difficult to believe). Because the authors assume a mature structure in their analysis, it is essential their assumption is correct. A very simple way to do this would be to perform a Western blot with the commercially available antibody 2H2 or attempt to enhance infection of their virus preparations on K562 with a prM-reactive antibody.

*** As detailed in my first review, the Lok lab has a wonderful paper detailing the numerous forms of DENV2 that may exist once incubated physiological temperatures. This complicates the assumptions of the analysis here (which assume a mature structure). In response to this suggestion, the authors indicate their results are incapable of detecting the contribution this type of heterogeneity. Because this is essentially a demonstration of a method, it is critical to distinguish between the lack of this type of heterogeneity in the sample and clean data (less plausible) or an inability to discern movement among non-mature T=3 structures (which makes the method less powerful). The biology fit the model? Or is the model defining the biology?

*** How quantitative is the mass spec? The authors have focused on the sensitivity of the instrumentation, but have not addressed how they estimate the relative efficiency of peptide generation by pepsin among particle types and symmetry environments. There are no standards used for this approach.

Reviewer #2 (Remarks to the Author):

The authors have addressed the majority of my concerns; and I appreciate the additional controls that were performed.

I do still feel that the title stating "proteome-wide" is too broad and does not accurately reflect the content of the paper. "Conformational changes of the dengue virus structural proteins in intact viruses..." or something along those lines would be more suitable.

Reviewer #3 (Remarks to the Author):

The new section describing HDX levels in known neutralizing antibody epitopes (p. 22) is useful. The authors should also discuss briefly how the broadly conserved class of E-dimer interface antibody epitopes is affected (or not) by HDX. Antibodies that bind E-dimer

interface epitopes have shown recent promise and therapeutic potential against dengue (and Zika) virus (see Rouvinski et al & Rey (2015); Barba-Spaeth et al & Rey (2016), both in Nature). It would be useful to discuss these epitopes, even if there is no major HDX signal in this class of epitopes.

All other concerns have been adequately addressed.

Response to Reviewers

Reviewer #1 (Remarks to the Author):

**** The PAGE gel of purified virus is a nice approach to demonstrate the relative cleavage of prM in their preparations, provided the authors are confident they have correctly assigned each viral gene product to each band. Evidence to support this is lacking. The conclusion that DENV2 NGC virus produced by culture in C636 insect cells is efficiently cleaved by furin would be unprecedented (and difficult to believe). Because the authors assume a mature structure in their analysis, it is essential their assumption is correct. A very simple way to do this would be to perform a Western blot with the commercially available antibody 2H2 or attempt to enhance infection of their virus preparations on K562 with a prM-reactive antibody.*

To address this concern, we have demonstrated that our viral preparations contained very little prM as we used the DENV2 (NGC) and DENV1 (PVP159) virus prepared under identical conditions detailed in Fibriansah et al^{1,2}, in the laboratory of our collaborator and co-author Dr. Sheemei Lok. The preparation protocol from insect C6/36 cells was under identical conditions as that used in Fibriansah et al^{1,2}. In the revised supplementary information, we have included PAGE results showing the amount of prM protein (20 kDa) in our mature virus samples compared to an immature virus preparation (S. Fig 1A). Our SDS PAGE analysis of mature DENV showed a barely observable faint prM-protein band, whereas the immature DENV samples showed a clear distinct intense prM band. It is consistent with our cryo-EM observation that the sample used for HDXMS measurements contained only a few immature particles (S. Fig 1B). Two immature virus particles, characterized by their distinct morphology, are highlighted amidst a vast majority of mature virus particles. A micrograph of the immature preparation is provided alongside for reference. We thank the reviewer for suggesting a Western Blot with a prM-specific 2H2 antibody. Since Western Blotting using 2H2 antibodies is a highly sensitive method, the prM bands of both the mature and immature virus preparations have been confirmed (S. Fig. 1A). Consistent with the Coomassie stained gel, the intensity of the prM band in Western blot is much lower in our mature viral samples compared to the immature virus preparation. We have included a detailed description of the experimental steps in the material and methods section in P. 24 of the manuscript.

In Supplementary Fig. 1C, we confirmed the identity of each of the bands in PAGE using mass spectrometry. Gel extraction of each of the bands was carried out and digested with trypsin followed by mass spectrometry. The trypsin-digested peptides identified are in red and unequivocally assigned the bands to C, E and M- proteins respectively.

Experimental evidence from SDS PAGE, Western blot, mass spectrometry and the cryo-EM micrographs consistently indicated that the DENV samples used in our HDXMS studies contained only a very low amount of immature DENV contaminants which would not alter our conclusions. The standard deviation of our HDXMS measurements from triplicate experiments is within 0.3 Deuterons across all peptides. The significance threshold for differences in deuterium exchange upon temperature-dependent expansion is 0.5 Deuterons and any contributions of such low amounts of immature virus would be well within this 0.5 deuteron significance threshold. The low amounts of immature viral

particles do not interfere with the distinct temperature-dependent increases in HDXMS in DENV1 and 2. Both DENV1 and DENV2 contain equivalent low amounts of immature viral particle, but both serotype and strains show distinct increases in HDXMS that directly correlates with their unique temperature-dependent serotype/strain-specific expansion profiles. This provides further evidence that our significance threshold accounts for the low amounts of immature viral particle in our DENV1/2 samples.

Other published reports have also shown low levels of immature virus in mature virus samples purified from infected insect C6/36 cells. To quote Fibriansah et al ², *“The virus preparation contained little or no prM, indicating low levels of contamination by immature virus, as determined by Coomassie blue-stained SDS-PAGE”*. In another study, Kuhn et al ³ – ‘Structure of Dengue Virus: Implications for Flavivirus Organization, Maturation, and Fusion’ also report that this method of preparation of mature virus particle in C6/36 insect cells contained very low amounts of immature virus particles. To quote the authors of that study: *“Although preparations of dengue-2 from mosquito cells are often reported to contain large amounts of uncleaved prM, our preparations contained little or no prM as determined by gel electrophoresis followed by staining with Coomassie Brilliant Blue (data not shown)”*. Our experimental conditions indicate an efficient cleavage of immature virus particles, the reasons for which are unclear and beyond the scope of this study. There is also a clear distinction on the preparation protocol for mature and immature particles in C6/36 cells. This low level of immature virus in such mature virus preparations is not unprecedented and indicates a high efficiency of cleavage to generate mature viral particles under identical virus preparation methods in C6/36 cells. Nevertheless, we have confirmed the negligible amounts of immature viral particle in our samples by multiple methods and updated the Supplementary information section.

**** As detailed in my first review, the Lok lab has a wonderful paper detailing the numerous forms of DENV2 that may exist once incubated physiological temperatures. This complicates the assumptions of the analysis here (which assume a mature structure). In response to this suggestion, the authors indicate their results are incapable of detecting the contribution this type of heterogeneity. Because this is essentially a demonstration of a method, it is critical to distinguish between the lack of this type of heterogeneity in the sample and clean data (less plausible) or an inability to discern movement among non-mature T=3 structures (which makes the method less powerful). The biology fit the model? Or is the model defining the biology?*

Our HDX measurements are calculated based on centroid values of isotopic envelopes showing a Normal/ Gaussian distribution averaged over the entire ensemble of particles. Any heterogeneity will be apparent if present in large proportions in the sample and if the heterogeneity showed sharply different deuterium exchange profiles and kinetics. A small component of immature viral particles in the mature virus preparation did not significantly alter centroid values as it would be undetectable within the Normal/Gaussian deuterium exchange isotopic envelope profile. The effect of immature virus, at these low levels, on HDXMS measurements were insignificant and within the standard error of our measurements (~0.29D). This is further confirmed by the distinct serotype/strain-specific effects of temperature-dependent expansion between DENV1 and DENV2, where our results show, despite having the same low extent of immature viral particle in the sample, that there are distinct differences in

HDXMS which forms the basis for this manuscript. We have also additionally confirmed that there is no evidence for heterogeneity from our HDXMS spectra observed (in terms of multimodal distributions) in over 100 HDXMS experimental runs among all different conditions. This high reproducibility in the absence of such heterogeneity in our HDXMS measurements allows us to explain and look into the reason in more detail below.

Fibriansah *et al.*, indeed highlight 4 distinct conformations of DENV2 incubated at 37 °C. The ability to identify unique deuterium exchange profiles for each of the conformations and to resolve deuterium exchange from the global average is based on the assumption that the distinct conformations identified by cryo-EM snapshot measurements are static and not interchangeable in the timescale of deuterium exchange measurements in solution (D=1 min). Our HDXMS experiments suggest that the conformations observed by cryo-EM are interchangeable. Alternatively, if we consider that these conformations are non-interchangeable, the extent of deuterium exchange at D=1 min would have to be significantly different to be able to resolve these distinct patterns of exchange. This would be observable as multimodal deuterium exchange profiles. We do not observe any such characteristic multimodal deuterium exchange profiles in solution in any of the peptides in all C, E and M proteins under all conditions examined. These point more to the likelihood of the conformations being interchangeable in solution. Indeed, this is the power and value of alternative methods to examine the same biophysical phenomenon.

Further, deuterium exchange measurements are averaged across multiple viral particles and 180 copies of the constituent proteins within each viral particle. It is unclear if all 180 E-protein copies in single viral particles are uniform or show any conformational heterogeneity in solution.

Another aspect is the difference in information obtained from these two techniques. Cryo-EM captures a snapshot of conformations at the instant of flash freezing and offers a window into all possible discrete structures at the given instant of the experiment. These measurements do not preclude the interconvertibility of these conformations in solution with time. In contrast, HDXMS measures the exchange all particles in solution across an extended time of measurement, in our experiments that corresponds to a 1 min time of exchange. The measurement captures the ensemble average of all conformations in solution and rapidly interchanging conformations will be averaged out due to the ensemble behavior of the viral particles in solution.

**** How quantitative is the mass spec? The authors have focused on the sensitivity of the instrumentation, but have not addressed how they estimate the relative efficiency of peptide generation by pepsin among particle types and symmetry environments. There are no standards used for this approach.*

Any differences in the relative efficiency of pepsin cleavage of proteins in particle types or symmetry environments are removed by stringent denaturation of viral particles with a combination of denaturant and acidic pH. Treatment of DENV particles with 1.5M Guanidinium hydrochloride and 0.25M TCEP at pH 2.5 ensures effective denaturation of quaternary contacts prior to pepsin digestion. Under these harsh denaturing conditions, the virus particles would undergo extensive denaturation⁴ resulting in all viral protein (C, E and M) units on the virus particles being denatured and resulting in complete proteolysis. Pepsin proteolysis of these denatured viral proteins would thus be independent of the symmetry environments and particle types.

Reviewer #2 (Remarks to the Author):

The authors have addressed the majority of my concerns; and I appreciate the additional controls that were performed.

I do still feel that the title stating "proteome-wide" is too broad and does not accurately reflect the content of the paper. "Conformational changes of the dengue virus structural proteins in intact viruses..." or something along those lines would be more suitable.

We agree and thank the reviewer for pointing out. We have amended the title to "Conformational changes in intact dengue virus reveal serotype-specific expansion"

Reviewer #3 (Remarks to the Author):

The new section describing HDX levels in known neutralizing antibody epitopes (p. 22) is useful. The authors should also discuss briefly how the broadly conserved class of E-dimer interface antibody epitopes is affected (or not) by HDX. Antibodies that bind E-dimer interface epitopes have shown recent promise and therapeutic potential against dengue (and Zika) virus (see Rouvinski et al & Rey (2015); Barba-Spaeth et al & Rey (2016), both in Nature). It would be useful to discuss these epitopes, even if there there is no major HDX signal in this class of epitopes. All other concerns have been adequately addressed.

We thank the reviewer for their suggestion to describe our HDX results at the E-dimer dependent epitopes. We have included a section in the discussion (p.22 and 23) describing temperature-specific changes in deuterium exchange with expansion of DENV1 and 2 at these epitope sites. Further, the possible implications of DENV1 and DENV2 expansion on this class of antibody binding have also been briefly discussed.

REFERENCES

1. Fibriansah G, *et al.* Cryo-EM structure of an antibody that neutralizes dengue virus type 2 by locking E protein dimers. *Science* **349**, 88-91 (2015).
2. Fibriansah G, *et al.* Structural Changes in Dengue Virus When Exposed to a Temperature of 37 degrees C. *J Virol* **87**, 7585-7592 (2013).
3. Kuhn RJ, *et al.* Structure of dengue virus: Implications for flavivirus organization, maturation, and fusion. *Cell* **108**, 717-725 (2002).
4. Roberts PL, Lloyd D. Virus inactivation by protein denaturants used in affinity chromatography. *Biologicals* **35**, 343-347 (2007).

REVIEWERS' COMMENTS:

Reviewer #1 (Remarks to the Author):

While I am surprised by the data presented in Figure S1, the authors have done a fantastic job responding to my comments about prM contamination of their preparation. Very convincing. This will make a very nice Nature Communications paper.

Response to Reviewers:

Reviewer #1 (Remarks to the Author):

While I am surprised by the data presented in Figure S1, the authors have done a fantastic job responding to my comments about prM contamination of their preparation. Very convincing. This will make a very nice Nature Communications paper.

-We thank the reviewer for this feedback.